# CountsDiff: A diffusion model on the natural numbers for generation and imputation of count-based data

## Abstract

Diffusion models have excelled at generative tasks for both continuous and token-based domains, but their application to discrete ordinal data remains underdeveloped. We present *CountsDiff*, a diffusion framework designed to natively model distributions on the natural numbers. CountsDiff extends the Blackout diffusion framework by simplifying its formulation through a direct parameterization in terms of a survival probability schedule and an explicit loss weighting. This introduces flexibility through design parameters with direct analogues in existing diffusion modeling frameworks. Beyond this reparameterization, CountsDiff introduces features from modern diffusion models, previously absent in counts-based domains, including continuous-time training, classifier-free guidance, and churn/remasking reverse dynamics that allow non-monotone reverse trajectories. We propose an initial instantiation of CountsDiff and validate it on natural image datasets (CIFAR-10, CelebA), demonstrating the benefits of the proposed design space and that the framework scales to complex, high-dimensional data domains. We then highlight biological count assays as a natural use case, evaluating CountsDiff on single-cell RNA-seq imputation in a fetal cell and heart cell atlas. Remarkably, we find that even this simple instantiation matches or surpasses the performance of a state-of-the-art discrete generative model and leading RNA-seq imputation methods, while leaving substantial headroom for further gains through optimized design choices in future work.

## 1 Introduction

Diffusion modeling (Sohl-Dickstein et al., 2015; Ho et al., 2020) is the state-of-the-art generative modeling framework, producing diverse and high-quality samples across various domains, including but not limited to images (Saharia et al., 2022; BlackForestLabs, 2025), audio (Lemercier et al., 2024), videos (Ho et al., 2022), and proteins (Abramson et al., 2024; Watson et al., 2023). These models define a forward noising process that iteratively corrupts data samples, transforming the data distribution into an easily sampled "noise" distribution, and then learn to reverse the noising process. Diffusion models have been well studied and developed in both continuous (Ho et al., 2020; Song et al., 2020a;b) and discrete, categorical (Austin et al., 2021; Hoogeboom et al., 2021; Campbell et al., 2022) data types, including for popular use cases such as images and tokenized text. However, data from biological assays such as whole-genome sequencing, RNA sequencing (including single-cell RNA sequencing; scRNA-seq), ATAC-seq (including single-cell ATAC-seq), and metagenomic read counts are direct measurements of abundance in the form of natural numbers. Like the reals, the natural numbers are an unbounded ordered set. However, the natural numbers are clearly discrete. Given this, both approaches must be adapted to count-based data. One can either (1) relax the natural numbers to the reals and train a continuous diffusion model (Kotelnikov et al., 2023; Jolicoeur-Martineau et al., 2024) or (2) treat each number up to some maximum as an independent class and train a discrete diffusion model. Both of these solutions present potential pitfalls. Training a continuous diffusion model optimizes over the space of real-valued distributions, only to quantize at inference time, which, as we illustrate using a simple toy dataset, can be ineffective. On the other hand, the categorical adaptation requires a distinct category for each possible value, which

quickly becomes computationally expensive as the maximum value increases and ignores the natural ordering of numerical data.

To address these challenges, we introduce *CountsDiff*, a modern diffusion model that operates on the set of natural numbers $\mathbb{N}_0 := \{0, 1, 2, \dots\}$. CountsDiff builds on the theoretical underpinnings of Blackout Diffusion (Santos et al., 2023), but, for greater clarity and generality, reparameterizes the forward process in terms of a survival probability $p(t)$, stabilizes the outputs via random rounding, and introduces analogs to the complete toolkit of contemporary diffusion models, including continuous-time training and sampling (Campbell et al., 2022), weighted objectives (Kingma & Gao, 2023), guidance (Dhariwal & Nichol, 2021; Ho & Salimans, 2022; Nisonoff et al., 2024), and churn/remasking (Song et al., 2020a; Karras et al., 2022; Wang et al., 2025). Furthermore, we evaluate CountsDiff across three settings. First, we use synthetic count data to illustrate the limitations of existing diffusion models on $\mathbb{N}_0$ compared to CountsDiff. Next, we test CountsDiff on natural images (CIFAR-10 (Krizhevsky et al., 2009) and CelebA (Liu et al., 2015)) to stress its ability to scale to high-dimensional distributions and examine the relative effects of the design parameters we introduce: noise schedules, loss weighting, and attrition. Finally, we turn to single-cell RNA-seq imputation, benchmarking CountsDiff on fetal and heart cell atlases (Cao et al., 2020; Litviňuková et al., 2020), to demonstrate a natural application for our count-based model.

## 2 BACKGROUND AND NOTATION

### 2.1 GENERATIVE LATENT VARIABLE MODELS

Given data $\mathcal{D} = \{x^{(i)}\}_{i=1}^N$ sampled from an unknown distribution $P_{\text{data}}$, the goal of generative modeling is to learn a distribution $P_{\text{gen}}$ to approximate $P_{\text{data}}$, often by minimizing negative log-likelihood (NLL, see Appendix A.1). A common approach to this is latent variable modeling, where one samples from a simple distribution $P_{\text{noise}}$ (*e.g.* unit Gaussian) and learns a transformation to the data domain.

Diffusion models follow this paradigm by defining a sequence of forward corruption kernels $\{q(x_t \mid x_{t-1})\}_{t=1}^T$ that gradually transform samples from $\mathrm{x}_0 \in P_{\text{data}}$ into $\mathrm{x}_T \in P_{\text{noise}}$. By composition, this defines a forward process $q(x_{1:T} \mid x_0)$ that maps the data to "noise" *e.g.,* samples from an isotropic Gaussian. Generative modeling seeks to approximate the corresponding reverse kernels $q(x_{t-1} \mid x_t)$, which ideally invert the corruption at each step. For instance, as we detail in Appendix A.2, when the support $\mathcal{X} = \mathbb{R}$, *Gaussian diffusion models* (Ho et al., 2020) use Gaussian transitions as forward kernels.

### 2.2 DISCRETE DIFFUSION MODELS

When data are discrete categories, one can define a diffusion process over a finite support with $|\mathcal{X}| = N$ via forward transition kernel matrices $\{\boldsymbol{Q}_t\}_{t=1}^T$, where $[\boldsymbol{Q}_t]_{i,j} = q(x_t = j \mid x_{t-1} = i)$. By composition,

$$q(x_t \mid x_0 = i) = \boldsymbol{e}^{(i)} \bar{\boldsymbol{Q}}_t, \quad \bar{\boldsymbol{Q}}_t = \boldsymbol{Q}_1 \boldsymbol{Q}_2 \dots \boldsymbol{Q}_t.$$

This general formulation admits a variety of forward processes that converge to simple $P_{\text{noise}}$'s. A common choice is to choose a simple categorical distribution $\mathrm{Cat}(\mathcal{X}, F)$ with $F \in \Delta^{N-1}$ and define

$$\boldsymbol{Q}_t = (1 - \beta_t)\,\mathbf{I} + \beta_t\,\mathbf{1}F^\top,$$

with schedule $\{\beta_t\}$. This yields marginals $\bar{\boldsymbol{Q}}_t = \bar{\alpha}_t \mathbf{I} + (1 - \bar{\alpha}_t)\,\mathbf{1}F^\top$, where $\bar{\alpha}_t = \prod_{s=1}^t (1 - \beta_s)$. Two important special cases are uniform discrete diffusion (Hoogeboom et al., 2021) when $F = \frac{1}{N}\mathbf{1}$ and masked diffusion (Austin et al., 2021; Sahoo et al., 2024; Shi et al., 2024) when $F = \boldsymbol{e}^{(\text{mask})}$. In both cases, the model is trained to approximate the reverse kernels $q(x_{t-1} \mid x_t)$ (Austin et al., 2021). We will refer to these models operating on *finite* categorical spaces as "categorical diffusion models" to contrast with CountsDiff, which is also formally discrete diffusion.

These processes extend naturally to continuous time (Shi et al., 2024; Sahoo et al., 2024). As $T \to \infty$, cumulative transition matrices $\bar{\boldsymbol{Q}}(t)$ evolve according to the Kolmogorov forward equations (Norris, 1997):

$$\tfrac{d}{dt}\bar{\boldsymbol{Q}}(t) = \bar{\boldsymbol{Q}}(t)\,\boldsymbol{R}(t), \quad q(x_t \mid x_0 = i) = \boldsymbol{e}^{(i)}\bar{\boldsymbol{Q}}(t). \tag{1}$$

$\boldsymbol{R}(t)$ specifies infinitesimal transition rates via

$$q(x_{t+dt} = j \mid x_t = i) = \delta_{i,j} + \boldsymbol{R}_{i,j}(t)\, dt + o(dt),$$

where $\delta$ is the Kronecker delta and $o(dt)$ represents terms that tend to zero faster than $dt$. This continuous-time formulation via transition rates enables the extension of diffusion principles beyond finite categorical data, see Benton et al. (2024); Holderrieth et al. (2024).

### 2.2.1 DISCRETE DIFFUSION ON THE NATURAL NUMBERS

To model data on $\mathbb{N}_0$, a natural choice is the family of birth–death processes (Karlin & McGregor, 1957; Feller et al., 1971), in which each state can only increase by one with birth rate: $\boldsymbol{R}_{i,i+1}(t) = \lambda_i(t)$, decrease by one with death rate $\boldsymbol{R}_{i,i-1}(t) = \mu_i(t)$ or stay the same with rate $\boldsymbol{R}_{i,i}(t) = -(\lambda_i(t) + \mu_i(t))$. Explicitly,

$$\boldsymbol{R}_{i,j}(t) = \lambda_i(t)(\delta_{i+1,j} - \delta_{i,j}) + \mu_i(t)(\delta_{i-1,j} - \delta_{i,j}).$$

Santos et al. (2023) restricts this to a pure-death process with $\lambda_i(t) = 0$ and $\mu_i(t) = i$, which is an absorbing-state forward process (*e.g.* masked categorical diffusion), *i.e.,* where $P_{\text{noise}}$ is a Dirac delta. Solving the Kolmogorov forward equation 1 yields binomial marginals:

$$q(x_t \mid x_0) = \binom{x_0}{x_t} p(t)^{x_t}(1 - p(t))^{x_0 - x_t}, \quad p(t) = \mathrm{e}^{-t}.$$

The corresponding reverse process is a pure-birth process with maximum state $x_0$ with rates

$$\boldsymbol{R}_{i,j}^{(\text{rev})}(t) = (x_0 - i)\frac{\mathrm{e}^{-t}}{1 - \mathrm{e}^{-t}}(\delta_{i+1,j} - \delta_{i,j}). \tag{2}$$

Learning this reverse process amounts to predicting the number of remaining elements $y_t = x_0 - x_t$ given $(x_t, t)$. This can be optimized by minimizing $\sum_{x_0 \in \mathcal{D}} \mathbb{E}_t\big[(\mathrm{e}^{-t_{k-1}} - \mathrm{e}^{-t_k})(\hat{y}_t - y_t \log \hat{y}_t)\big]$, which is equivalent to minimizing the NLL (Santos et al., 2023).

## 3 METHODS

Herein, we formally introduce the CountsDiff framework, defining a forward process parameterized by a $p$-schedule, a weighted loss, a family of reverse processes parameterized by an attrition schedule, predictor-free guidance, and a stochastic rounding algorithm that prevents a failure mode of Blackout Diffusion.

### 3.1 COUNTSDIFF FORWARD PROCESS

For our forward process, we consider the following *inhomogeneous* pure-death process

$$\boldsymbol{R}_{i,j}^{(\text{fw})}(t) = i\mu(t)(\delta_{i-1,j} - \delta_{i,j}).^1 \tag{3}$$

We define a CountsDiff forward process with $p$-*schedule* $p(t)$ as a pure death process with transitions given by equation 3, choosing $\mu(t)$ such that the process has marginals

$$q(x_t \mid x_0) = \binom{x_0}{x_t} p(t)^{x_t}(1 - p(t))^{x_0 - x_t}, \tag{4}$$

and conditionals

$$q(x_t \mid x_s) = \binom{x_s}{x_t} \left(\frac{p(t)}{p(s)}\right)^{x_t} \left(1 - \left(\frac{p(t)}{p(s)}\right)\right)^{x_s - x_t}. \tag{5}$$

We have the following existence proposition, proven in Appendix B.1:

**Proposition 1.** *Given $p : [0,1] \to [0,1]$ differentiable, monotonically decreasing, and with endpoints $p(0) = 1$, $p(1) = 0$, there exists a CountsDiff forward process with $p$-schedule $p(t)$.*

This forward process is visualized in Figure 1. Blackout diffusion's forward process is a special case of CountsDiff's; see Appendix B.2. We note that when matching the signal-to-noise ratios (SNR) of CountsDiff and Gaussian diffusion, $p(t)$ is analogous to noise schedule $\bar{\alpha}_t$. This correspondence enables the adaptation of any noise schedule from Gaussian diffusion to CountsDiff; for this work, we propose $p(t) = \cos(\frac{\pi t}{2})^2$ from Nichol & Dhariwal (2021) (see Appendix B.3), which also has some theoretical advantages over the schedule in Blackout diffusion (see Appendix B.7).

---

[1]Blackout diffusion is a special case of this when $\mu(t) \equiv 1$.

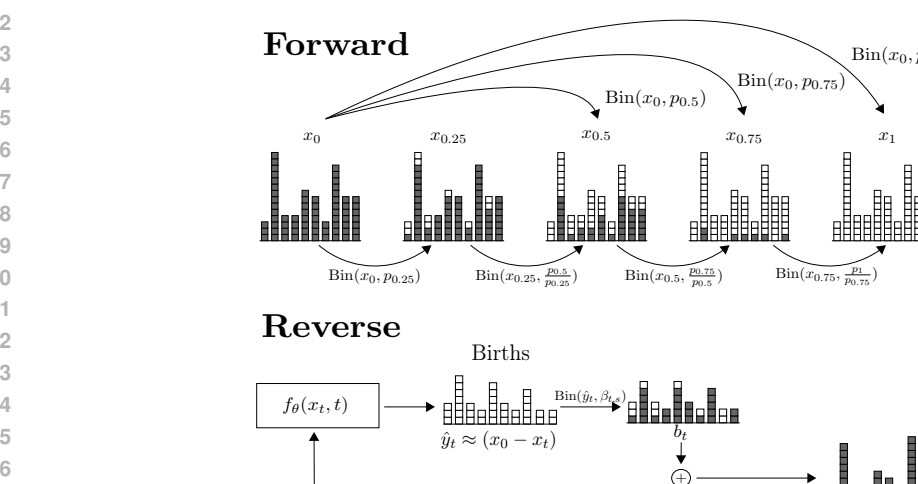

Figure 1: Visualization of CountsDiff's forward corruption process (top) and reverse sampling process (bottom). The top diagram depicts the progression of a $p$-schedule, a pure death process. The bottom shows a single step of the generalized, birth-death sampling process.

## 3.2 COUNTSDIFF OBJECTIVE

Our general objective takes the form

$$\mathbb{E}_{t\sim\phi}\left[w(t)(\hat{y}_t - y_t \log \hat{y}_t)\right], \qquad \hat{y}_t = [\text{NN}_\theta(x_t, t)]^+, \tag{6}$$

where $\text{NN}_\theta$ is a neural network, $w(t) > 0$ is a weighting term, and $[a]^+ := \max(a, 0)$.

When $w(t) = \frac{-p'(t)}{(1-p(t))\,\phi(t)}$, where $\phi(t)$ is the distribution $t$ is sampled from, the loss recovers the NLL. (see Appendix B.4). It is easy to see that since this objective is minimized pointwise by taking $\hat{y}_t = y$, for any $w(t) > 0$, the minimizer remains unchanged, and the choice of weight only influences our training dynamics. We refer to Kingma & Gao (2023) for a more in-depth discussion on the weighting of diffusion model losses and their correspondence with importance sampling.

For our cosine $p$-schedule $p(t) = \cos(\frac{\pi t}{2})^2$ we propose a weighting term $w(t) = \frac{\pi}{2}\sin(\pi t)$, which can be derived using two orthogonal heuristics: matching the sigmoid weighting commonly used in Gaussian diffusion, or by matching the form $w(t) = -p'(t)$ in Blackout diffusion. In fact, the choice of $-p'(t)$ also results in equation 6 corresponding to an exact NLL (see prop 3 in Appendix B.5). For a detailed discussion, see Appendix B.6.

## 3.3 COUNTSDIFF REVERSE PROCESS WITH ATTRITION

The reverse process (proof in Appendix B.8) corresponding to equation 3 ,

$$\boldsymbol{R}_{i,j}^{(\text{rev})} = (x_0 - i)\frac{p'(t)}{1-p(t)}(\delta_{i,j} - \delta_{i+1,j}), \tag{7}$$

generates a monotonic trajectory via a pure-birth process, a limitation analogous to the irreversible nature of unmasking in masked diffusion models. Discrete diffusion models overcome this through remasking (Wang et al., 2025). Analogously, we generalize the reverse process to allow *attrition*, a nonzero death rate compensated with births to preserve the binomial marginal:

**Proposition 2** (Reverse step with attrition). *Given $x_t$, a p-schedule $p(t)$, and an attrition rate $\sigma_{t,s} \in [0, \sigma_{t,s}^{\max}]$, where $\sigma_{t,s}^{\max} := \min(1, \frac{1-p(s)}{p(t)})$, let $\beta_{t,s} = \frac{p(s)-(1-\sigma_{t,s})p(t)}{1-p(t)}$. Then the following sampling procedure preserves the marginal distribution of $\mathrm{x}_s$ defined in equation 4:*

$$\mathrm{x}_s = \mathrm{n}_t + \mathrm{b}_t \qquad \mathrm{n}_t \sim \text{Bin}(x_t, 1 - \sigma_{t,s}), \quad \mathrm{b}_t \sim \text{Bin}(x_0 - x_t, \beta_{t,s}). \tag{8}$$

See Appendix B.9 for a proof of this proposition. Varying this $\sigma_{t,s}$, which is analogous to the churn parameter in Gaussian diffusion (Song et al., 2020a; Karras et al., 2022) and remasking in discrete diffusion (Wang et al., 2025), yields a family of birth-death processes. All elements of this family are valid given a trained model because the training procedure (Algorithm 1) depends only on the marginals. Then, given an attrition schedule $\sigma$, we can generate samples by iteratively performing equation 8, where $x_0 - x_t$ is replaced by the prediction $\hat{y}$ from a neural network trained to optimize equation 6. This sampling procedure is visualized in Figure 1 and described in Algorithm 2.

Both $\sigma_{t,s}^{\max}$ and $\beta_{t,s}$ as a function of $\sigma_{t,s}$ have the same form as their analogs in ReMDM (Wang et al., 2025), providing an interpretation of a birth as an *unmasking* event and a death as a *remasking* event. Thus, we borrow a remasking strategy from ReMDM as a starting point: ReMDM-rescale, which sets $\sigma_{t,s} = \eta_{\text{rescale}} \sigma_{t,s}^{\max}$, where $\eta_{\text{rescale}}$ is a tunable sampling hyperparameter.

### 3.4 GUIDANCE

Classifier-free guidance is a widely used technique in continuous and categorical diffusion models (Dhariwal & Nichol, 2021; Ho & Salimans, 2022), and was recently extended to discrete state spaces in continuous time by Nisonoff et al. (2024); Schiff et al. (2024); Li et al. (2024). We adapt the method of the Nisonoff et al. (2024) to enable guidance for diffusion models on the natural numbers. Formally, given a conditional reverse rate $\boldsymbol{R}_{i,j}^{(\text{rev})}(t \mid c)$ for class $c$ and its unconditional counterpart $\boldsymbol{R}_{i,j}^{(\text{rev})}(t)$, the guided rate $\boldsymbol{R}_{i,j}^{(\text{rev})}(t; \gamma \mid c)$ with strength $\gamma \geq 0$ is defined as

$$\boldsymbol{R}_{i,j}^{(\text{rev})}(t; \gamma \mid c) = \boldsymbol{R}_{i,j}^{(\text{rev})}(t \mid c)^\gamma \boldsymbol{R}_{i,j}^{(\text{rev})}(t)^{1-\gamma}.$$

For CountsDiff, this simplifies to

$$\hat{y}^{(\gamma)} = (\hat{y} \mid c)^\gamma \hat{y}^{1-\gamma},$$

where guided samples are obtained by substituting $\hat{y}^{(\gamma)}$ directly into the binomial reverse process. As in other diffusion frameworks, we implement predictor-free guidance by training a single neural network that outputs both conditional and unconditional predictions, achieved by randomly zeroing out class embeddings with probability $p_{\text{uncond}}$, typically set to $0.1$ or $0.2$.

### 3.5 ROUNDING

At inference time, rounding is necessary to convert the real-valued output of neural networks to predictions $\hat{y} \in \mathbb{N}_0$, as required by the reverse process. Naively rounding to the nearest integer causes mode collapse at 0 when $\hat{y} < 0.5$ (which occurs frequently for near-zero counts). Santos et al. (2023) and Chen & Zhou (2023) consider a scheme based on a Poisson approximation; however, this expression is an unfaithful approximation of the binomial distribution in this small $y$ setting. Instead, we adopt a randomized rounding scheme that preserves the expectation of $\hat{y}$ while keeping exact binomial draws, preventing 0-collapse (appendix E.1.3) in a principled manner.

$$\hat{y}_{\text{clipped}} = \lfloor \hat{y} \rfloor + \xi, \qquad \xi \sim \text{Bernoulli}(\hat{y} - \lfloor \hat{y} \rfloor).$$

### 3.6 ADAPTING COUNTSDIFF TO DATA IMPUTATION

We adapt CountsDiff to imputation using the RePaint algorithm (Lugmayr et al., 2022, Algorithm 1), originally developed for image inpainting. RePaint requires no retraining: after each reverse step during sampling, observed entries are reset to their noised ground-truth values, and only masked entries are resampled. This procedure has been successfully repurposed in other domains (*eg.* Forest Diffusion (Jolicoeur-Martineau et al., 2024)), and we adopt it here for biological count data.

## 4 EXPERIMENTS

We validate CountsDiff in three experimental settings. We begin with a small toy dataset of counts, comparing it with Gaussian and categorical diffusion. We follow with experiments on digital images (CIFAR-10 and CelebA (Krizhevsky et al., 2009; Liu et al., 2015)) to show that CountsDiff is capable of modeling complex, high-dimensional distributions, and to probe the relative effect of different

$p$-schedules, guidance, and reverse process dynamics in a visually interpretable setting with well-defined benchmarks. Finally, we propose scRNA-seq imputation as a natural real-world use-case for CountsDiff and benchmark it against existing imputation methods.

**Simulated counts**: To validate CountsDiff's strength in generating counts, we train three simple models on synthetic 10-dimensional sparse count vectors: a Gaussian diffusion model operating in log-space, a (categorical) masked diffusion model (MDLM (Sahoo et al., 2024), which is equivalent to ReMDM (Wang et al., 2025) with no remasking), and CountsDiff without attrition. Data are generated from negative binomial distributions with multiplicative size factors, yielding $\approx 50\%$ zeros and maximum counts near 50. Each model uses a small multi-layer perceptron (MLP) backbone with matching parameter counts. We evaluate sample quality using Maximum Mean Discrepancy (MMD) (Gretton et al., 2012) and sliced Wasserstein-1 distance (SWD), a scalable approximation of the Wasserstein-1 distance, using 100 random projections following (Bonneel et al., 2015). We also computed the MMD and Wasserstein-1 distance between the marginal distribution of each dimension. We also approximations of the distributions of a subset of the dimensions using Kernel Density Estimator (KDE) plots. Full distributional parameters and architecture details are in Appendix D.1.

**Natural images**: We train CountsDiff on CIFAR-10 and CelebA using U-Net architectures adapted from prior diffusion baselines (Song et al., 2020b), following the hyperparameters of Santos et al. (2023) wherever possible. We report three main experiments:

1. **Guidance.** We implement predictor-free guidance with $p_{\text{uncond}} = 0.1$ and evaluate conditional sampling across guidance scales. We measure FID/IS and inspect CIFAR-10 samples qualitatively.
2. **Reverse-process dynamics.** We introduce nonzero attrition during sampling and assess its effect on FID/IS, as well as any visual effects on images. To validate robustness, we illustrate this on both CIFAR-10 and CelebA.
3. **Quantitative results.** We compare the FI discrete schedule implied by (Santos et al., 2023), its continuous analog, and our proposed cosine schedule, and report the best-performing combinations of $p-$schedule, guidance, and attrition on 50k CIFAR-10 samples.

**Single cell RNA-Seq imputation**: We evaluate CountsDiff and a series of baselines on three imputation tasks on heart cell Litviňuková et al. (2020) and fetal cell Cao et al. (2020) atlases: 50% missing complete at random (MCAR) for fetal cells, a 25% low-biased missing not at random (MNAR) regime (low counts are masked out) for fetal cells, and 50% MCAR for heart cells. See Appendix A.3 for further discussion of missingness and imputation.

We preprocess our data to select a subset of genes that are commonly but differentially expressed across cells (see Appendix D.3.1). Imputation target sites are masked out and imputed using various baseline methods. These results are evaluated according to sample-level metrics[2] and log single-cell FID (log(scFID)), an adaptation of FID for scRNA-seq (Rizvi et al., 2025) as a distributional metric (implementation details in Appendix D.3.2). Experimental and training settings for all baselines are detailed in Appendix D.3.3. For each metric, we obtain standard error by resampling the generated values (50k data points for the fetus cell atlas and 20k for the heart cell atlas) ten times. See Appendiex D.3.4 for a brief discussion of our chosen metrics.

## 5 RESULTS

### 5.1 TOY EXAMPLE

Both CountsDiff and masked diffusion are easily able to learn the marginals of the toy distribution, with comparably low marginal MMD and Wasserstein-1 distances across all dimensions (Figure 2). Both also perform well on the joint MMD, indicating they capture the dominant modes well.

However, masked diffusion performs poorly on SWD, indicating it may be overfitting to outliers in the training set and is therefore more prone to generating excessive outliers/low-quality "hallucinations." We confirm this by noting that the variance in each dimension of samples generated by masked diffusion is far greater than the other two models and the real data (see Figure 2 and Appendix E.1). The higher joint-SWD relative to marginal Wasserstein-distance also indicates masked

---

[2]root mean-squared error (RMSE), bias, and Spearman's rank correlation are computed for each sample and averaged over the entire evaluation set.

diffusion is less effective in learning the correlations between dimensions; we observe this in the joint KDE plots between the first two dimensions (Figure 8).

Gaussian diffusion, on the other hand, was entirely unable to learn these sparse, discrete, ordinal data and suffered from extreme mode collapse. We were unable to match the performance of the discrete models, even by increasing model capacity and training time and varying learning rate.

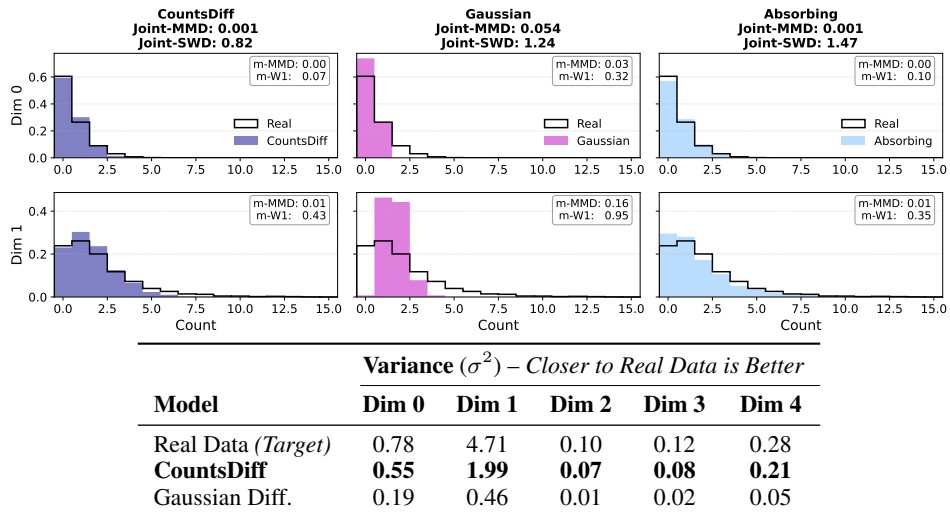

| | **Variance** ($\sigma^2$) – *Closer to Real Data is Better* | | | | |
|---|---|---|---|---|---|
| **Model** | **Dim 0** | **Dim 1** | **Dim 2** | **Dim 3** | **Dim 4** |
| Real Data *(Target)* | 0.78 | 4.71 | 0.10 | 0.12 | 0.28 |
| **CountsDiff** | **0.55** | **1.99** | **0.07** | **0.08** | **0.21** |
| Gaussian Diff. | 0.19 | 0.46 | 0.01 | 0.02 | 0.05 |
| Masked Diff. | 3.06 | 9.22 | 1.89 | 2.49 | 7.27 |

Figure 2: Histogram of model-generated samples versus ground truth (top), and variance statistics (bottom) for a subset of dimensions. Existing diffusion models exhibit failure cases even in a simple toy dataset: Gaussian diffusion suffers from mode collapse, while masked diffusion overfits outliers (inflated variance). Full results for all ten dimensions can be found in Appendix E.1.

## 5.2 NATURAL IMAGES

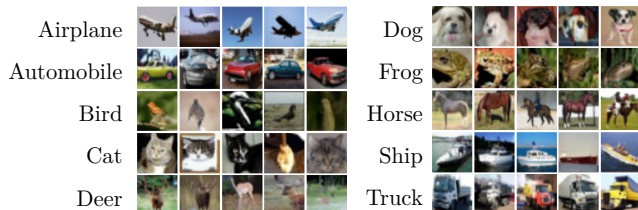

Figure 3: 5 images guided by each Cifar10 class sampled from CountsDiff with guidance scale 2.0 and $\eta_{\text{rescale}} = 0.005$.

**Guided Image Generation**: guidance performs as expected, enabling class-conditioned sampling for CIFAR-10 (Figure 3). Moderate levels of guidance also improve FID and IS (Table 1)

**Increasing attrition rate has a smoothing effect**: rough/noisy parts of an image can be interpreted as "overshooting" the correct value; allowing for these values to decrement manifests as smoothing. Taken to the extreme, we see dramatic oversmoothing, which results in a complete removal of texture and, eventually, perspective as $\eta_{\text{rescale}} \to 1$. See Figure 4 for CIFAR-10 and Figure 11 for CelebA.

### 5.2.1 QUANTITATIVE RESULTS

We find both moderate guidance and small, nonzero attrition to improve the FID and IS of samples generated by CountsDiff. Generalizing the FI p-schedule from Santos et al. (2023) to continuous time improves FID. Across hyperparameters, it seems that the FI noise schedule results in slightly better FID, while the cosine schedule results in slightly better IS, indicating the cosine schedule generates slightly higher fidelity samples at the expense of slightly poorer sample diversity. Notably, even this limited exploration of our extended design space with choices drawn directly from

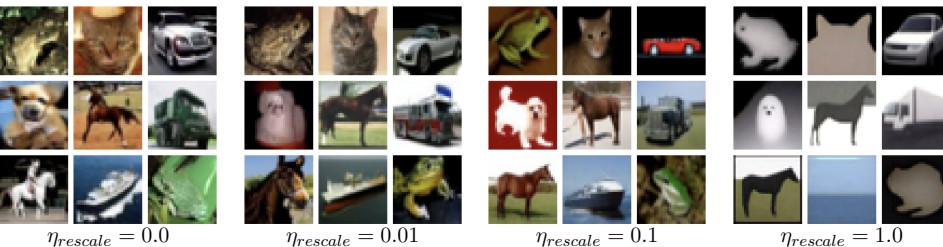

$\eta_{rescale} = 0.0$     $\eta_{rescale} = 0.01$     $\eta_{rescale} = 0.1$     $\eta_{rescale} = 1.0$

Figure 4: Nine images drawn from CountsDiff trained on CIFAR-10 with $\eta_{\text{rescale}}$ attrition schedule for varying levels of $\eta_{\text{rescale}}$

existing diffusion frameworks resulted in significant improvements in sample quality and diversity, underscoring the potential for the elucidated CountsDiff design space to enable rapid, systematic advances in count-based diffusion.

We also found training curves to be more stable for our cosine noise schedule (See Appendix E.2.1), consistent with the intention of this schedule in Gaussian diffusion. For more extensive ablations on $\gamma$ and attrition schedules see Appendix E.2.2, E.2.3. For results on CelebA see Appendix E.3.

Table 1: FID and IS of 50k images sampled with from CountsDiff trained on CIFAR-10 for a selected set of sampling hyperparameters. FI Discrete with unconditional generation and no attrition is equivalent to Blackout Diffusion

| $p$-**schedule** | $\gamma$ | attrition schedule | **FID** $\downarrow$ | **IS** $\uparrow$ |
|---|---|---|---|---|
| FI Discrete (Blackout) | unconditional | none | 5.726 | $9.124 \pm 0.049$ |
| FI Continuous | unconditional | none | 5.439 | $9.088 \pm 0.156$ |
| FI Continuous | 1.0 | $\eta_{\text{rescale}} = 0.01$ | **5.198** | $9.640 \pm 0.167$ |
| FI Continuous | 2.0 | $\eta_{\text{rescale}} = 0.02$ | 9.707 | $9.765 \pm 0.112$ |
| Cosine Continuous | unconditional | none | 5.756 | $9.287 \pm 0.175$ |
| Cosine Continuous | 1.0 | $\eta_{\text{rescale}} = 0.01$ | 5.261 | $9.853 \pm 0.083$ |
| Cosine Continuous | 2.0 | $\eta_{\text{rescale}} = 0.02$ | 11.546 | $\mathbf{9.929 \pm 0.128}$ |

## 5.3 SCRNA-SEQ IMPUTATION

Our imputation results for the MCAR and MNAR scenarios on the fetus cell atlas (Cao et al., 2020) are shown below in Table 2 and Table 3. Further results on the heart cell atlas (Litviňuková et al., 2020) can be found in Table 13 in Appendix E.4. On occasion, we observe a catastrophic collapse of scIDPM for some samples (fewer than 10), with the model imputing counts greater than $10^6$ across several genes of the same sample. We report results both including (scIDPMs 1-sample, 5-sample) and excluding (scIDPMs filtered) these samples. In multiple imputation, this instability in scIDPMs generation was difficult to resolve due to mean ensembling, resulting in worse sample quality and poorer evaluation metrics.

In the MCAR scenario, CountsDiff achieves the highest performance in scFID, outperforms ReMDM in RMSE for single imputation, and is indistinguishable from ReMDM in bias. Impressively, we outperform mean imputation in RMSE. We have equally strong results in the low-biased MNAR scenario, achieving the best RMSE and bias, and beaten slightly by only ReMDM in scFID and Spearman correlation. Notably, CountsDiff is the least-biased method in the MNAR scenario.

Despite ReMDM being one of the state-of-the-art generative models for discrete data modalities, this early implementation of CountsDiff demonstrates comparable performance across the scRNA-seq imputation tasks. We also observe that ReMDM has higher RMSE and substantially higher RMSE standard error compared to CountsDiff, reflecting its tendency to over-sample outliers. This tendency is particularly undesirable in scientific settings, where outliers are precisely the observations used to infer signal; over-sampling is therefore likely to generate spurious findings and false conclusions. Furthermore, CountsDiff has about half as many parameters (one-fourth in the heart

Table 2: Benchmarking results on scRNA-seq imputation on human fetus cell atlas with 50% MCAR. Metrics are reported as mean (standard error)

| Model Method | Spearman ↑ | RMSE↓ | Bias | log(scFID) ↓ |
|---|---|---|---|---|
| RAW | N/A | 1.888(0.032) | $-1.317(0.002)$ | $-2.344(0.015)$ |
| Mean imputation | 0.047(0.001) | 1.320(0.065) | 0.001(0.003) | $-5.149(0.021)$ |
| Conditional Mean | 0.199(0.001) | 1.092(0.039) | 0.002(0.003) | $-6.214(0.046)$ |
| MAGIC | **0.202(0.008)** | 1.907(0.062) | $-1.314(0.004)$ | $-2.366(0.020)$ |
| scIDPMs, 1-sample | 0.088(0.001) | 2.594(0.059) | 0.971(0.002) | $-2.694(0.019)$ |
| scIDPMs, 5-samples | 0.083(0.001) | $> 10^3$ | $> 10^3$ | $-3.037(0.008)$ |
| GAIN | 0.021(0.001) | 51.163(0.106) | 20.122(0.058) | 1.400(0.001) |
| Hi-VAE | 0.000(0.001) | 1.719(0.032) | 0.621(0.003) | $-0.712(0.003)$ |
| Forest-Diffusion | 0.003(0.001) | 19.694(0.101) | 5.426(0.011) | $-0.886(0.007)$ |
| ReMDM, 1-sample | 0.109(0.001) | 1.769(0.138) | $-\mathbf{0.003}(\mathbf{0.001})$ | $-9.101(0.001)$ |
| ReMDM, 5-samples | 0.123(0.001) | *1.196 (0.083)* | $-\mathbf{0.003}(\mathbf{0.001})$ | $-8.460(0.001)$ |
| CountsDiff, 1-sample | 0.094(0.001) | 1.401(0.047) | *0.004(0.002)* | $-\mathbf{9.253}(\mathbf{0.071})$ |
| CountsDiff, 5-samples | 0.115(0.001) | **1.195(0.060)** | *0.004(0.001)* | $-7.600(0.073)$ |

Table 3: Benchmarking results on scRNA-seq imputation on human fetus cell atlas with 25% low-biased missingness (MNAR). Metrics are reported as mean (standard error)

| Model Method | Spearman ↑ | RMSE↓ | Bias | log(scFID) ↓ |
|---|---|---|---|---|
| RAW | N/A | 0.998(0.0001) | $-0.994(0.0002)$ | $-5.272(0.017)$ |
| Mean imputation | 0.248(0.006) | 0.898(0.0001) | $-0.895(0.0002)$ | $-5.726(0.012)$ |
| Conditional Mean | 0.425(0.003) | 0.469(0.001) | 0.285(0.001) | $-5.761(0.009)$ |
| MAGIC | 0.143(0.039) | 0.997(0.0001) | $-0.994(0.0002)$ | $-5.273(0.018)$ |
| scIDPMs, 1-sample | $-0.369(0.001)$ | $> 10^3$ | $> 10^3$ | $-1.010(0.007)$ |
| scIDPMs, filtered | 0.285(0.006) | 2.195(0.022) | 1.209(0.002) | $-2.198(0.020)$ |
| scIDPMs, 5-sample | 0.108(0.0012) | $> 10^3$ | $> 10^3$ | $-3.649(0.015)$ |
| GAIN | 0.279(0.004) | 52.333(0.0907) | 27.953(0.065) | $-1.663(0.006)$ |
| Hi-VAE | 0.006(0.003) | 0.890(0.0003) | $-0.3010(0.0004)$ | $-1.941(0.005)$ |
| Forest-Diffusion | 0.059(0.001) | 4.393(0.145) | 1.400(0.002) | $-2.023(0.015)$ |
| ReMDM, 1-sample | **0.468(0.004)** | 1.020(0.138) | 0.314(0.002) | **-6.688 (0.022)** |
| ReMDM, 5-sample | 0.374(0.004) | 0.636(0.025) | 0.312(0.002) | $-6.477(0.024)$ |
| CountsDiff, 1-sample | 0.418(0.003) | 0.871(0.009) | **0.299(0.002)** | $-6.409(0.025)$ |
| CountsDiff, 5-sample | 0.355(0.002) | **0.580(0.006)** | **0.300(0.002)** | $-6.144(0.023)$ |

cell atlas task) as ReMDM, due to ReMDM's output layer size depending on the max count. With further optimization in $p$-scheduling, loss weighting, and attrition scheduling beyond the scope of this work, there is substantial room for empirical improvement to the CountsDiff models.

## 6 RELATED WORK

### 6.1 GENERATIVE MODELS

Our work is most closely related to Blackout Diffusion (Santos et al., 2023), which can be interpreted as a special case of CountsDiff with no guidance, fixed $p$-schedule and loss weighting, and no sampling with attrition. Santos et al. (2023) also prove the NLL objective and validity of the reverse process in their special case.

JUMP (Chen & Zhou, 2023) models positive, real-valued data by projecting it into counts ($z_0 \sim$ poisson($\lambda x_0$)), then noises through a binomial thinning of $z_0$, resulting in a similar noising process, parametrized by $\alpha_t$. Their loss objective, derived from the ELBO, resembles equation 6 but with a different predictive target and constant loss weighting. For natively count-based data, Chen & Zhou (2023) also propose Binomial-JUMP, which can be interpreted as another special case of CountsDiff with constant weights, no guidance, and no attrition, and uses the Poisson sampling scheme mentioned in section 3.5. JUMP's primary advantage lies in its ability to handle continuous non-negative data, which is outside the scope of the present work. The underlying noising and denoising

resembles Blackout Diffusion and has a similarly limited design space. JUMP and CountsDiff are complementary approaches, and CountsDiff's improvements on modeling counts (extended design space, continuous-time formulation, and exact loss) can be readily extended to non-negative reals using JUMP's Poisson data randomization trick (Appendix A.5).

We would also like to point the reader towards relevant works in Gaussian and categorical diffusion that can help inform design choices of the CountsDiff design space. In particular, (Karras et al., 2022) and (Kingma & Gao, 2023) explore noise schedules and loss weighting in Gaussian diffusion; Wang et al. (2025) introduces remasking for masked discrete diffusion, which is analogous to attrition; and Sahoo et al. (2025) introduces a framework to bridge Gaussian diffusion with discrete diffusion in order to more easily transfer design choices.

### 6.2 SCRNA-SEQ IMPUTATION

Due to the high sparsity and missingness of scRNA-seq data (as discussed in Appendix A.3), imputation of scRNA-seq data is an important but challenging problem. Various methods have been proposed to address this issue. Data diffusion and manifold learning methods, such as MAGIC (Van Dijk et al., 2018), attempt to build a similarity graph across similar cells and average a cell's expression profile with those of its closest neighbors. Generative methods for scRNA-seq imputation include adaptations of GANs (MisGAN (Li et al., 2019), GAIN (Yoon et al., 2018), CT-GAN (Xu et al., 2019)), VAEs (HI-VAE (Nazabal et al., 2020), scVAE (Grønbech et al., 2020), AdImpute (Xu et al., 2021)), and more recently, continuous diffusion generative models such as scIDPMs (Zhang & Liu, 2024). Methods such as Forest-Diffusion, which is capable of imputing tabular data through diffusion gradient-boosted trees (Jolicoeur-Martineau et al., 2024), can also be adapted to this task. Other diffusion models on scRNA-seq exist, including scDiffusion (Luo et al., 2024), Squidiff (He et al., 2024), and scDesign3 (Song et al., 2024), but these models are not intended for imputation. Rather, they are designed as purely generative models capable of generating synthetic data or making downstream predictions.

## 7 DISCUSSION

In this paper, we introduced CountsDiff, a diffusion framework designed to handle discrete ordinal data using birth/death processes as the noising and denoising mechanisms. Our main contribution is an elucidated and deconvolved design space, where each design parameter, noise schedule, loss weighting, reverse-process modifications, and guidance, has a direct and interpretable analogue in modern continuous and categorical diffusion models. This framing both extends and clarifies the design space of Blackout Diffusion (Santos et al., 2023). Concretely, we unlocked continuous-time training for Santos et al. (2023), reparameterized the model with a more intuitive $p$-schedule, introduced a principled loss weighting, derived attrition as the counterpart to churn/remasking, and incorporate classifier-free guidance.

We proposed principled starting points for each of these new design parameters, demonstrating how our unified design space enables seamless transfer across diffusion families. Through experiments on a range of applications, from image generation to scRNA-seq imputation, we demonstrated that this initial instantiation of CountsDiff matches the performance of a state-of-the-art discrete diffusion model while avoiding a key failure case in discrete, ordinal regimes. Further, CountsDiff also outperformed specialized scRNA-seq imputation methods across multiple metrics.

Our work has yielded promising results for scRNA-seq imputation, and we expect further hyperparameter optimization and task-specific training adaptations to unlock the potential of the CountsDiff framework for this and other large-scale biology applications with count data, such as ATAC-Seq imputation, perturbation effect prediction, and single-cell foundation modeling.

## 8 REPRODUCIBILITY AND CODE AVAILABILITY

A functional implementation and demo of CountsDiff, along with code necessary to reproduce the results in this work will be provided to reviewers. The code will be open sourced upon publication.

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

## A EXTENDED BACKGROUND

### A.1 KULLBACK-LEIBLER DIVERGENCE EQUIVALENCE WITH NEGATIVE LOG LIKELIHOOD

Because the data distribution is unknown and unknowable, requiring that $P_{\text{gen}} \approx P_{\text{data}}$ is ill-defined. This problem is commonly addressed by approximating the objective using Monte-Carlo sampling. For example, if the error is quantified by the Kullback-Leibler divergence (Kullback & Leibler, 1951):

$$D_{\text{KL}}(P_{\text{data}}|P_{\text{gen}}) = \mathbb{E}\left[\log \frac{P_{\text{data}}(x)}{P_{\text{gen}}(x)}\right],$$

we can approximate it via

$$\mathbb{E}\left[\log \frac{P_{\text{data}}(x)}{P_{\text{gen}}(x)}\right] \approx \sum_{x \in D} \log(P_{\text{data}}(x)) - \log(P_{\text{gen}}(x)),$$

which is minimized with respect to $P_{\text{gen}}$ when the Negative Log Likelihood (NLL) of the data with respect to $P_{\text{gen}}$, $NLL = -\sum_{x \in D} \log(P_{\text{gen}}(x))$, is minimized, since the first term is constant with respect to $P_{\text{gen}}$.

### A.2 GAUSSIAN DIFFUSION MODELS

Gaussian diffusion models are diffusion models where the forward kernels are Gaussian transitions

$$q(\mathbf{x}_t|\mathbf{x}_{t-1}) = \mathcal{N}(\sqrt{1 - \beta_t}\mathbf{x}_{t-1}, \beta_t\mathbf{I}),$$

where the sequence $\{\beta_t\}_{t=1}^T$ is a monotonic *variance schedule* with $\beta_0 = 0$ and $\beta_T = 1$. The Gaussian diffusion forward process can be rewritten in the following closed form:

$$\mathbf{x}_t = \sqrt{\bar{\alpha}_t}\mathbf{x}_0 + \sqrt{1 - \bar{\alpha}_t}\epsilon, \epsilon \sim \mathcal{N}(0, \mathbf{I}), \tag{9}$$

with $\bar{\alpha}_t = \Pi_{s=1}^t(1 - \beta_t)$, resulting in $P_{\text{noise}}(z|\mathbf{x})$ $q(\mathbf{x}_T|\mathbf{x}_0) = \mathcal{N}(0, \mathbf{I})$. Optimizing $p_\theta(\mathbf{x}_{t-1}|\mathbf{x}_t)$ to minimize the NLL of the observed $\mathbf{x}_{t-1}$ can be reduced to predicting the added noise $\epsilon$, the original signal $\mathbf{x}_0$, or some hybrid of the two. Though these objectives are equivalent up to reweighting of the loss objective, their empirical performance can vary (Kingma & Gao, 2023). This process and the corresponding objectives can also naturally be extended to the continuous time domain by taking the limit as $T \to \infty$ and $\Delta t \to 0$. The forward and reverse processes become stochastic differential equations (SDEs) (Song et al., 2020b), but the marginals can still be written in closed form, similar to equation 9:

$$\mathbf{x}_t = \sqrt{\alpha(t)}\mathbf{x}_0 + \sqrt{1 - \alpha(t)}\epsilon, \epsilon \sim \mathcal{N}(0, \mathbf{I}), \tag{10}$$

where $\alpha(t)$ is commonly referred to as a *noise schedule*, and is a continuous monotonic function of $t$. Although the resulting training objectives are nearly identical, the continuous extension allows for more flexibility at the generation stage: one can sample using numerical stochastic differential equations / ordinary differential equations solvers (Hartman, 2002) (ODE), or if they choose to discretize the reverse SDE, they are no longer bound to a specific number of time steps. Due to the similarity in the training objectives and the Gaussianity of the marginals $q(x_t|x_0)$ in these continuous extensions, we will consider them a subclass of Gaussian diffusion models, namely *continuous-time* (as opposed to *discrete-time*) Gaussian diffusion models.

### A.3 DATA MISSINGNESS

The standard theory for missing data depends on the notion of "missing at random" (MAR, (Rubin, 1976; Seaman et al., 2013; Schafer, 1997)). There are three types of missingness mechanisms: (i) *missing completely at random* (MCAR), when the process determining missingness is assumed to be independent of (the values of) the variables; (ii) *missing at random* (MAR), when the missingness mechanism depends only on the observed variables, and (iii) *missing not at random* (MNAR), when the missingness depends on both observed and unobserved (missing) variables (Little & Rubin, 2020).

More formally, suppose there is some ground truth vector of counts $\mathbf{x}^{\text{true}}$, and some binary mask $\mathbf{o}$, and we observe $\mathbf{x}^{\text{obs}} = \mathbf{x}^{\text{true}} \cdot \mathbf{o}$.

The MCAR assumption is simply that each position $o_i$ of $o$ is distributed according to

$$o_i \overset{i.i.d}{\sim} \text{Bernoulli}(1-d)$$

for some dropout probability $d$.

The MAR assumption is that $o \perp\!\!\!\perp x^{\text{true}}$, and MNAR covers all remaining cases. In particular, we are interested in the setting where for each $o_i$, we have

$$o_i \sim \text{Bernoulli}(1-d_i),$$

where $d_i$ is larger for smaller values of $x_i$. This form of MNAR is relevant for scRNA-seq imputation tasks, as missingness can be induced by low read counts (Qiu, 2020).

## A.4 IMPUTATION

Missingness can be addressed with either *single* or *multiple imputation*. Multiple imputation (MI) (Rubin, 2004) is a widely studied method (Sterne et al., 2009; Hayati Rezvan et al., 2015) that uses the distribution of observed data to estimate a set of likely values for missing data. By contrast, single imputation generates only a single point.

Several methods have been developed to handle data missingness. Early methods include complete case analysis, in which samples with missing data are simply removed from the dataset, and mean imputation, where missing values are filled with the per-variable mean across all (or a subset satisfying a specific condition) of the observed data points. Early machine learning methods include random forest imputation (Hastie et al., 2009; Stekhoven & Bühlmann, 2012; Shah et al., 2014), which recursively splits the data via a predictor from known samples to estimate unobserved values.

More recently, generative models, including variational autoencoders (VAEs) (Kingma & Welling, 2014), generative adversarial networks (GANs) (Goodfellow et al., 2014), and diffusion models Sohl-Dickstein et al. (2015); Ho et al. (2020), have been explored for data imputation. These methods rely on the principle that generative models are capable of learning the underlying data-generating distributions. In Burda et al. (2015); Nazabal et al. (2020); Roskams-Hieter et al. (2023), the authors extend VAEs, initially replacing missing values with zeros and training a neural network to predict these values. Similarly, Li et al. (2019); Yoon et al. (2018); Xu et al. (2019) use GANs, setting up a game between a generator $G$ that generates *both* observed and imputed values and a discriminator $D$ that decides whether a particular data point was imputed or not. Diffusion models can be designed for imputation (Tashiro et al., 2021; Alcaraz & Strodthoff, 2022) or be adapted for imputation through algorithms such as RePaint (Lugmayr et al., 2022), which passes the noised ground truth at each time step of denoising to fix the imputation target sites (Jolicoeur-Martineau et al., 2024), or methods such as DiffPuter (Zhang et al., 2025), which uses the Expectation-Maximization algorithm to guide a diffusion model to fill in missingness.

### A.4.1 MULTIPLE IMPUTATION

As described in Huque et al. (2018), there are two general approaches for imputing incomplete data: (a) joint modeling (JM) (Hughes et al., 2014) and (b) fully conditional specification (FCS) or multiple imputation using chained equations (MICE) (Van Buuren et al., 2006; Van Buuren & Groothuis-Oudshoorn, 2011). In JM a multivariate distribution of the missing data is sampled using Markov chain Monte Carlo (MCMC) (Gilks et al., 1995). In cases where this multivariate distribution is suitable for the data, this method is appealing. FCS employs a set of conditional densities, one for each partially observed variable, and performs imputation in a variable-by-variable manner. This is done by starting from an initial imputation and then imputing by iterating a few times (usually 10-20) over the conditional densities.

### A.5 EXTENDING TO CONTINUOUS DATA

While our work focuses on modeling count data (which allows us to preserve the exact NLL), the Poisson-based data randomization trick from Chen & Zhou (2023) can be combined with CountsDiff via the following procedure:

1. Nonnegative inputs $x_0$ are mapped to latent counts via $z_0 \sim \text{Poisson}(\lambda x_0)$, $\lambda \geq 1$.

2. CountsDiff is applied directly to model the distribution of $z_0$

3. Generated samples are divided by $\lambda$ at inference time. Chen & Zhou (2023) show that the original distribution of $x_0$ is recovered as $\lambda \to \infty$.

This simple procedure would extend the benefits of CountsDiff (guidance, schedule design, loss weighting, and attrition) to JUMP and therefore provide a principled way to model continuous, non-negative domains. The continuous time formulation also in principle unlocks fast ODE/SDE solvers (Ren et al., 2025) for JUMP. We note that this would not be a *strict* generalization of JUMP, as the model would operate directly in the latent counts space, as opposed to combining the Poisson randomization with binomial thickening/thinning at each forward and reverse step, and the training target would be $z_0 - z_t$ as opposed to $x_0$ in JUMP.

## B    PROOFS AND DERIVATIONS

### B.1    PROOF OF PROPOSITION 1

We restate the proposition here for clarity:

Given $p : [0,1] \to [0,1]$ differentiable, monotonically decreasing, and with endpoints $p(0) = 1$, $p(1) = 0$, there exists a CountsDiff forward process with $p$-schedule $p(t)$

*Proof.* Fix $x_0 \in \mathbb{N}_0$ and consider $x_0$ independent, two-state (0/1) time-inhomogeneous Markov processes

$$\mathrm{y}_t^{(m)} \in \{0,1\}, \qquad m = 1, \dots, x_0,$$

each with transition rates $\boldsymbol{R}^{(\mathrm{fw})}(t) = \begin{bmatrix} 0 & 0 \\ \mu(t) & -\mu(t) \end{bmatrix}$. Let $\mathrm{x}_t = \sum_{m=1}^{x_0} \mathrm{y}_t^{(m)}$ be the number of ones at time $t$; then $\mathrm{x}_t$ is governed by the pure-death process in equation 3.

For a single particle, consider the survival probability $\mathbb{P}(\mathrm{y}_t = 1 \mid \mathrm{y}_0 = 1)$. The Kolmogorov forward equation 1 then yields

$$\frac{d}{dt}\mathbb{P}(\mathrm{y}_t = 1 \mid \mathrm{y}_0 = 1) = -\mu(t)\mathbb{P}(\mathrm{y}_t = 1 \mid \mathrm{y}_0 = 1), \qquad P(\mathrm{y}_0 = 1 \mid \mathrm{y}_0 = 1) = 1,$$

which is a separable ODE with solution $\mathbb{P}(\mathrm{y}_t = 1 \mid \mathrm{y}_0 = 1) = \exp\left(-\int_0^t \mu(u)\, du\right)$. Let $\mu(t) := -\frac{p'(t)}{p(t)}$ for $t \in (0,1]$, and $\mu(0) = 0$. This choice is always valid since $p$ is differentiable and nonincreasing, and is positive at $t \in (0,1]$. Furthermore, the choice of value of $\mu$ at $t = 0$ does not influence the dynamics of the process, as $\exp\left(-\int_0^0 \mu(u)\, du\right)$ always equals 1, as the integrand is form 0 to 0. Thus we can by inserting our ansatz derive

$$\mathbb{P}(\mathrm{y}_t = 1 \mid \mathrm{y}_0 = 1) = \exp\left(\int_0^t \frac{p'(u)}{p(u)}\, du\right) = \exp\left(\log p(t) - \log p(0)\right) = \frac{p(t)}{p(0)} = p(t).$$

Thus $\mathrm{y}_t^{(m)} \sim \text{Bernoulli}(p(t))$ i.i.d. across $m$. Since the sum of i.i.d Bernoulli's is Binomial, we have

$$\mathbb{P}(\mathrm{x}_t = x_t \mid \mathrm{x}_0 = x_0) = \binom{x_0}{x_t} p(t)^{x_t} (1 - p(t))^{x_0 - x_t},$$

which is equation 4.

For $0 \le s < t \le 1$, we have by Bayes theorem

$$\mathbb{P}(\mathrm{y}_t = 1 \mid \mathrm{y}_s = 1) = \frac{\mathbb{P}(\mathrm{y}_s = 1 \mid \mathrm{y}_t = 1)\mathbb{P}(\mathrm{y}_t = 1 \mid \mathrm{y}_0 = 1)}{\mathbb{P}(\mathrm{y}_s = 1 \mid \mathrm{y}_0 = 1)} = \frac{1 \cdot p(t)}{p(s)} = \frac{p(t)}{p(s)},$$

where $\mathbb{P}(\mathrm{y}_s = 1 \mid \mathrm{y}_t = 1) = 1$ because a pure-death process alive at $t$ has to have been alive at $s$. Conditioning on $X_s = x_s$, the $x_s$ survivors evolve independently, so

$$(X_t \mid X_s = x_s) \sim \text{Bin}\left(x_s, \frac{p(t)}{p(s)}\right),$$

which gives the binomial conditionals in equation 5. $\qquad\square$

### B.2 REPARAMETERIZATION OF BLACKOUT DIFFUSION TIME SCHEDULE AS A $p$-SCHEDULE

Santos et al. (2023) work with a pure death-noising process with constant individual death rate $\mu \equiv 1$. As such, in order to adjust their corruption process for a constant decay in Fisher Information (FI), they define the following time schedule:

$$t_k = -\log\left[\sigma\left(\text{Logit}(1 - e^{-t_T}) + \frac{k-1}{T-1}[\text{Logit}(e^{-t_T}) - \text{Logit}(1 - e^{-t_T})]\right)\right], \quad k = 1, \ldots T. \tag{11}$$

However, note that the $\log$ term undoes the $\exp$ in their $p$-schedule, so this schedule is effectively a workaround to allow for a non-exponential $p$-schedule

$$p_k = \sigma\left(\text{Logit}(1 - e^{-t_T}) + \frac{k-1}{T-1}[\text{Logit}(e^{-t_T}) - \text{Logit}(1 - e^{-t_T})]\right), \quad k = 1, \ldots T.$$

However, using the time-inhomogeneous pure-death process in equation 3, we can bypass the time schedule trick, so that configurability lies directly in $p$ space, which we find to be a more intuitive schedule that better matches existing diffusion literature. For greater consistency with continuous-time diffusion frameworks, we have also rescaled the time steps to be on the closed unit interval. As a concrete example, the $p$-schedule from blackout diffusion becomes

$$p(t) = \sigma(\text{Logit}(1 - p_{min}) + t \cdot [\text{Logit}(p_{min}) - \text{Logit}(1 - p_{min})]), \quad t \in [0, 1],$$

where $p_{min}$ defines the values at the endpoints and is set to $e^{-15}$. Note that despite the extension of $p(t)$ to a continuous function, sampling $T$ uniform values from 0 to 1 exactly recovers the original formulation.

### B.3 EQUIVALENCE OF $p$-SCHEDULE AND GAUSSIAN DIFFUSION NOISE SCHEDULE

Our $p$-schedule is inspired by the cosine noise schedule in Nichol & Dhariwal (2021), where their noise schedule takes the following form:

$$\bar{\alpha}(t) = \cos\left(\frac{t/T + s}{1 + s}\frac{\pi}{2}\right)^2.$$

Taking the most canonical form, with $s = 0$ and $T = 1$, we have

$$\bar{\alpha}(t) = \cos\left(\frac{t\pi}{2}\right)^2.$$

To find the CountsDiff analog, we match the signal-to-noise ratio (SNR) of the cosine noise schedule in Gaussian diffusion with the SNR of the pure death process and solve for $p(t)$. In Gaussian diffusion, this takes the form

$$\text{SNR}(t) = \frac{\bar{\alpha}_t}{1 - \bar{\alpha}_t} = \frac{\cos^2(\pi t/2)}{\sin^2(\pi t/2)}.$$

In our pure-death process with $p$-schedule $p(t)$, the SNR can be expressed as $\frac{p(t)}{1-p(t)}$. Using the same signal to noise definition $\text{SNR}(t) = \frac{\mathbb{E}[x_t]^2}{\text{Var}(x_t)}$ as for the gaussian case. Thus for the independent Bernoullis underlying our pure-death process, we have

$$\text{SNR}(t) = \frac{p(t)^2}{p(t)(1 - p(t))} = \frac{p(t)}{1 - p(t)}.$$

Clearly then, $\bar{\alpha}_t$ is analogous to $p(t)$, so choosing $p(t) = \cos\left(\frac{t\pi}{2}\right)^2$ is a sensible choice. A similar exercise can be done for any $\bar{\alpha}$ schedule in Gaussian Diffusion, yielding a CountsDiff equivalent.

### B.4 DERIVING TRAINING OBJECTIVE; PROOF OF PROPOSITION 3

Santos et al. (2023) derives a continuous time loss function by taking the Kullback-Leibler divergence between Bernoulli distributions corresponding to instantaneous transitions of the ground truth

reverse process $\boldsymbol{R}_{i,i+1}^{(\text{rev})}\Delta t$, and the reverse process induced by the model predictions $\kappa_\theta(t)\Delta t$, at time $t$, where $\Delta t$ is an infinitesimal time differential. This corresponds to the negative log-likelihood and in our notation takes the form

$$\text{NLL}(t) = \boldsymbol{R}_{x_t,x_t+1}^{(\text{rev})}\Delta t \log \frac{\boldsymbol{R}_{x_t,x_t+1}^{(\text{rev})}\Delta t}{\kappa_\theta(t)\Delta t} + (1 - \boldsymbol{R}_{x_t,x_t+1}^{(\text{rev})}\Delta t) \log \frac{1 - \boldsymbol{R}_{x_t,x_t+1}^{(\text{rev})}\Delta t}{1 - \kappa_\theta(t)\Delta t}.$$

We simplify by splitting the logarithm of the fraction in the second term, and Taylor expanding, yielding

$$-(1 - \boldsymbol{R}_{x_t,x_t+1}^{(\text{rev})}\Delta t) \log (1 - \kappa_\theta(t)\Delta t) = \kappa_\theta(t)\Delta t + \mathcal{O}(\Delta t^2).$$

Collecting terms that do not depend on our model parameters $\theta$ into the "constant" $C(t)$, and omitting the higher order terms of $\Delta t$ gives us the representation

$$\text{NLL}(t) = \Delta t \left( \kappa_\theta(t) - \boldsymbol{R}_{x_t,x_t+1}^{(\text{rev})} \log \kappa_\theta(t) \right) + C(t).$$

For the full negative log-likelihood, we take the integral over all times and get

$$\int_0^1 \left( \kappa_\theta(t) - \boldsymbol{R}_{x_t,x_t+1}^{(\text{rev})} \log \kappa_\theta(t) \right) dt + C,$$

where C is the integral of all the $\theta$-independent terms, which is omitted in the sequel. Now, we can multiply and divide by any probability density function $\phi : [0,1] \to [0,1]$ over $t$

$$\int_0^1 \frac{1}{\phi(t)} \left( \kappa_\theta(t) - \boldsymbol{R}_{x_t,x_t+1}^{(\text{rev})} \log \kappa_\theta(t) \right) \phi(t) dt,$$

which allows us to approximate this integral with the usual one-sample Monte Carlo estimate with $t \sim \phi(t)$, resulting in the objective

$$\frac{1}{\phi(t)} \left( \kappa_\theta(t) - \boldsymbol{R}_{x_t,x_t+1}^{(\text{rev})} \log \kappa_\theta(t) \right), \quad t \sim \phi(t).$$

Notice from equation 7, that, given $t$ and $x_t$, only unknown element of the reverse rate is $(x_0 - x_t) =: y_t$. Consequently, we train a neural network $\text{NN}_\theta(x_t, t)$ to output $\hat{y}(t, x_t) \approx y_t$.

Then, we have $\kappa_\theta(t) = \hat{y}(t, x_t) \frac{p'(t)}{1-p(t)}$, which together with equation 7 yields the objective

$$-\frac{p'(s)}{\phi(t)(1 - p(s))} \left( \hat{y}(x_t, t) - (x_0 - x_t) \log \left( -\hat{y}(x_t, x_t) \frac{p'(s)}{1 - p(s)} \right) \right).$$

Finally, dropping the $\hat{y}$-independent term, we get the objective

$$w(t) \left( \hat{y}(x_t, t) - (x_0 - x_t) \log (\hat{y}(x_t, t)) \right), \quad w(t) = -\frac{p'(s)}{\phi(t)(1 - p(s))},$$

### B.5 NLL FOR CHOSEN WEIGHTING FUNCTION

**Proposition 3.** *Let $p : [0,1] \to (0,1)$ be a continuously differentiable, strictly decreasing p-schedule of a CountsDiff forward process $C_p$. Define the training weight*

$$w(t) = -p'(t),$$

*and consider the objective*

$$\mathcal{L}(\theta) = \mathbb{E}_{t \sim \text{Unif}(0,1)} \left[ w(t) \left( \hat{y}_t - y_t \log \hat{y}_t \right) \right], \qquad y_t = x_0 - x_t.$$

*Then there exists a CountsDiff process $C_q$ with forward schedule*

$$q(t) = 1 - \exp\left( p(1) - p(t) \right),$$

*such that $\mathcal{L}(\theta)$ coincides with the continuous-time negative log-likelihood of $C_q$ under uniform time sampling, up to a $\theta$-independent scaling and additive constant.*

*Moreover, there exists $F : [0,1] \to [0,1]$ (the CDF of a density $\phi$) such that*

$$p(u) = q\big( F^{-1}(u) \big) \quad \text{for all } u \in [0,1].$$

*Consequently, for any uniform grid $0 = u_0 < \cdots < u_K = 1$, running the forward or reverse sampler of $C_p$ at times $\{u_k\}$ is equivalent to running the sampler of $C_q$ at the warped grid $\{t_k\}$ with $t_k = F^{-1}(u_k)$.*

*Proof.* To prove the statement, we note that

$$\frac{q'(t)}{1 - q(t)} = \frac{\exp(p(1) - p(t))p'(t)}{\exp(p(1) - p(t))} = p'(t)$$

Therefore, we have

$$\mathbb{E}_{t \sim \text{Unif}(0,1)}\big[ - p'(t) \left(\hat{y}_t - y_t \log \hat{y}_t\right)\big] = \mathbb{E}_{t \sim \text{Unif}(0,1)}\big[ - \frac{q'(t)}{1 - q(t)} \left(\hat{y}_t - y_t \log \hat{y}_t\right)\big],$$

and for any valid importance sampling density $\phi$, this is equal to

$$\mathbb{E}_{t \sim \phi}\big[ - \frac{q'(t)}{\phi(t)(1 - q(t))} \left(\hat{y}_t - y_t \log \hat{y}_t\right)\big],$$

which is the exact negative log-likelihood for $C_q$. Then, to show the final part of the proposition it suffices to take $\phi(t) \propto \frac{d}{dt}(p^{-1}(q(t)))$

$\square$

### B.6 MOTIVATING WEIGHTING FUNCTION

Our first method of motivating is from the weighting term in blackout diffusion, $w(t_k) = (t_k - t_{k-1})e^{-t_k}$. As we take the limit $t_{k-1} \to t_k$, we approach our continuous case, and we get the weight $w(t) = |p'(t)dt|$, where $p(t) = e^{-t}$.

In our case, we have a uniform time schedule, $dt$ is constant, so we simply reweight by $w(t) = |p'(t)| = \frac{\pi}{2}\sin(\pi t)$. This is an intuitive choice, since time derivative of $p(t)$ can be thought of (informally) as the rate of information decrease, and a steeper decrease corresponds to a more difficult task to undo. Thus, it is sensible to weight by the magnitude $p'(t)$.

An alternate way to motivate this weighting using the sigmoid weighting function commonly used in Gaussian diffusion (Kingma et al., 2021) . When the training task is $\epsilon$-prediction, the sigmoid weight, defined as a function of the log-SNR $\ell(t)$, is $w(\ell) = \sigma(b - \ell)$, where $\sigma(x) = \frac{1}{1+e^{-x}}$ is the sigmoid function and $b$ is a chosen constant. When the training task is $x_0$-prediction (which is equivalent to $\epsilon$-prediction with additional reweighting), the sigmoid weighting $\hat{w}(\ell) = e^b \sigma(\ell - b)$. Since the log SNR takes the form $\ell(t) = \ln(\frac{p(t)}{1-p(t)})$, plugging in $\ell$ and $b = 0$ into the two sigmoid weightings above, we get that

$$w(\ell(t)) = \sigma(-\ln(\frac{p(t)}{1 - p(t)})) = \frac{1}{1 + \frac{p(t)}{1-p(t)}} = 1 - p(t),$$

and

$$\hat{w}(\ell(t)) = \sigma(\ln(\frac{p(t)}{1 - p(t)})) = \frac{1}{1 + \frac{1-p(t)}{p(t)}} = p(t).$$

Heuristically, $(x_0 - x_t)$ is an interpolation of predicting the initial state $x_0$ and predicting the step-wise additive noise $\epsilon$. In fact, taking a log-space interpolation between $w$ and $\hat{w}$:

$$
\begin{aligned}
w(\ell)^{1/2}\hat{w}(\ell)^{1/2} &= \sqrt{p(t)(1 - p(t))} \\
&= \sqrt{\cos^2(\pi t/2)\sin^2(\pi t/2)} \\
&= \cos(\pi t/2)\sin(\pi t/2) \\
&= \frac{1}{2}\sin(\pi t),
\end{aligned}
$$

matching our $w(t)$ up to a constant factor of $\pi$.

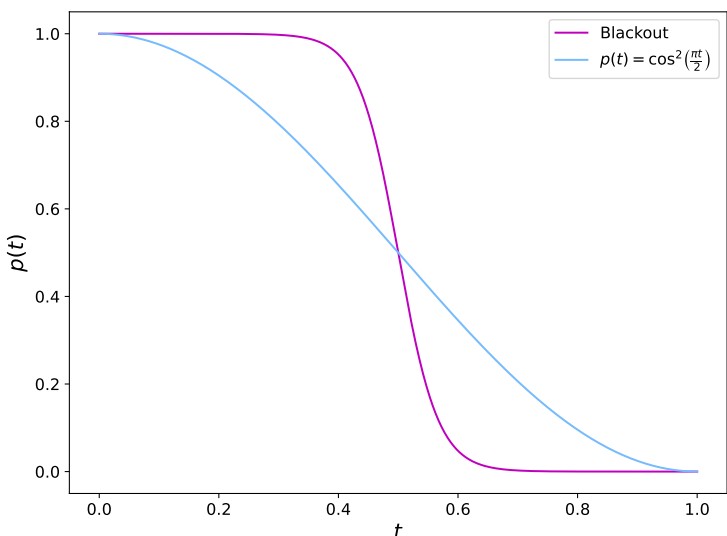

Figure 5: Converted $p-$schedule from Blackout Diffusion (see B.2) versus cosine $p$-schedule

### B.7 COMPARING PROPOSED $p$-SCHEDULE AND WEIGHTING WITH BLACKOUT DIFFUSION

As was the case with early linear noise schedules in Gaussian Diffusion, the exponential $p-$schedule described in Blackout Diffusion has potentially undesirable properties 0 and 1, where $p(t)$ is almost completely flat. The cosine schedule, on the other hand, decreases more gradually (see Figure 5). As a result, the corresponding weighting function for Blackout diffusion puts substantially more emphasis on time steps near 0.5, and close to no emphasis on those near the endpoints, effectively reducing the batch size, while our proposed weighting attributes a non-negligible weight to nearly all time points (Figure 6). These properties of the $p$-schedules and corresponding weighting schedules

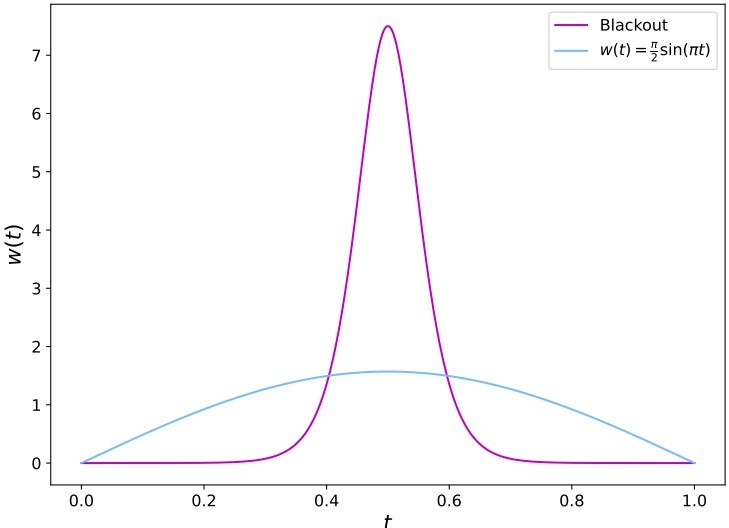

Figure 6: Weights from Blackout Diffusion versus proposed $p$-schedule

explain the improvement in training stability shown in Figure 10, and may be a factor in the more stable inception scores of samples generated by CountsDiff when trained with the cosine $p$-schedule.

## B.8 REVERSE PROCESS DERIVATION

We here construct the form of the rates $\boldsymbol{R}_{i,j}^{(\mathrm{rev})}(t)$ for the reverse process. Given the form of the binomial marginals $q(x_t \mid x_0)$ in equation 4, we can construct the reverse rate matrix by equating the forward and reverse rates between states $i$ and $i+1$

$$\boldsymbol{R}_{i,i+1}^{(\mathrm{rev})}(s)q(x_s = i|x_0) = \boldsymbol{R}_{i+1,i}^{(\mathrm{fw})}(t)q(x_s = i + 1 \mid x_0).$$

We then have the rate of an instantaneous transition from $i$ to $i+1$ as

$$
\begin{aligned}
\boldsymbol{R}_{i,i+1}^{(\mathrm{rev})}(s) &= \boldsymbol{R}_{i+1,i}^{(\mathrm{fw})}(t)\frac{q(x_s = i + 1 \mid x_0)}{q(x_s = i \mid x_0)} \\
&= (i+1)\mu(s)\frac{q(x_s = i + 1 \mid x_0)}{q(x_s = i \mid x_0)} \\
&= (i+1)\mu(s)\frac{\binom{x_0}{i+1}p(s)^{i+1}(1 - p(s))^{x_0-(i+1)})}{\binom{x_0}{i}p(s)^i(1 - p(s))^{x_0-i}} \\
&= (i+1)\mu(s)\frac{x_0!}{(x_0-(i+1))!(i+1)!}\frac{(x_0-i)!i!}{x_0!}\frac{p(s)}{1 - p(s)} \\
&= (i+1)\mu(s)\frac{(x_0-i)}{i+1}\frac{p(s)}{1 - p(s)} \\
&= (x_0-i)\mu(s)\frac{p(s)}{1 - p(s)} \\
&= -(x_0-i)\frac{p'(s)}{1 - p(s)},
\end{aligned}
$$

where in the final step we have inserted the explicit solution of $\mu(s) = -\frac{d}{ds}p(s)/p(s)$ expressed as a function of $p(t)$. This is an application of Bayes' theorem, but a more theoretical operator algebra-based treatment yields an analogous result in Appendix A of Santos et al. (2023). Since the reverse process is a pure birth process, the only allowed instantaneous transfers are between $i$ and $i+1$, and $i$ staying in state. Thus $\boldsymbol{R}_{i,i}^{(\mathrm{rev})}(s) = -\boldsymbol{R}_{i,i+1}^{(\mathrm{rev})}(s)$, $\boldsymbol{R}_{i,j}^{(\mathrm{rev})}(s) = 0$ otherwise. This yields the full formulation in equation 2.

## B.9 PROOF OF PROPOSITION 2

We restate the proposition for clarity:

Given $x_t$, a $p$-schedule $p(t)$, and an attrition rate $\sigma_{t,s} \in [0, \sigma_{t,s}^{\max}]$, where $\sigma_{t,s}^{\max} := \min(1, \frac{1-p(s)}{p(t)})$, let $\beta_{t,s} = \frac{p(s)-(1-\sigma_{t,s})p(t)}{1-p(t)}$. Then the following sampling procedure preserves the marginal distribution of $\mathrm{x}_s$ according to equation 4:

$$\mathrm{x}_s = \mathrm{n}_t + \mathrm{b}_t \qquad \mathrm{n}_t \sim \mathrm{Bin}(x_t, 1 - \sigma_{t,s}), \quad \mathrm{b}_t \sim \mathrm{Bin}(x_0 - x_t, \beta_{t,s}),$$

*Proof.* In order to prove the proposition, we need to show that if

$$q(x_t \mid x_0) = \binom{x_0}{x_t} p(t)^{x_t} (1 - p(t))^{x_0-x_t},$$

then we have for the $\mathrm{x}_s$ as sampled in the proposition statement that

$$q(x_s \mid x_0) = \binom{x_0}{x_s} p(s)^{x_s} (1 - p(s))^{x_0-x_s}$$

As with B.1, we will model $\mathrm{x}_t$ and $\mathrm{x}_s$ as the sum of $x_0$ independent, two-state Markov processes. Then, the sampling procedure proposed in equation 8 is equivalent to

$$(\mathrm{y}_s^{(m)} = 1|\mathrm{y}_t^{(m)} = 1) = 1 - \sigma_{t,s}, \qquad (\mathrm{y}_s^{(m)} = 1|\mathrm{y}_t^{(m)} = 0) = \beta_{t,s}.$$

Then, at time $t$, since $\mathbb{P}(\mathrm{y}_t^{(m)} = 1|\mathrm{y}_0 = 1) = p_t$, we have

$$\begin{aligned}
\mathbb{P}(\mathrm{y}_s^{(m)} = 1|\mathrm{y}_0 = 1) &= (1 - \sigma_{t,s})p(t) + \beta_{t,s}(1 - p(t)) \\
&= (1 - \sigma_{t,s})p(t) + p(s) - (1 - \sigma_{t,s})p(t) \\
&= p(s),
\end{aligned}$$

where for the second equality we have inserted the form of $\beta_{t,s} = \frac{p(s)-(1-\sigma_{t,s})p(t)}{1-p(t)}$ from the proposition statement. Thus we can conclude that $\mathrm{x}_s$ has the marginal binomial distribution we set out to prove.

To determine the range of validity for $\sigma_{t,s}$, we test the edge cases

$$\beta_{t,s} \leq 1 \implies \sigma_{t,s} \leq \frac{1 - p(s)}{p(t)}$$

$$\sigma_{t,s} \geq 0$$

One can easily check that the lower bound on $\beta_{t,s}$ does not impose an additional lower bound on $\sigma_{t,s}$.

Thus with our assumed attrition rate $\sigma_{t,s} \in [0, \sigma_{t,s}^{\max}]$, where $\sigma_{t,s}^{\max} := \min(1, \frac{1-p(s)}{p(t)})$, validity for $\beta_{t,s}, \sigma_{t,s}$ is guaranteed. $\qquad\square$

## C  ALGORITHMS

The training algorithm, Algorithm 1, largely aligns with that of Blackout Diffusion. The sampling algorithm, including our contributions, is outlined in Algorithm 2.

---

**Algorithm 1:** Training CountsDiff

---

1  **while** *not converged* **do**
2      Draw $x_0 \sim \mathcal{D}$ from the training set;
3      Sample $t \sim \mathrm{Unif}([0, 1])$;
4      Sample $x_t \sim \mathrm{Bin}(x_0, p(t))$ element-wise;
5      Predict: $\hat{y}_t \leftarrow \mathrm{NN}_\theta(x_t, t)$;
6      Compute: $y_t \leftarrow x_0 - x_t$;
7      Compute: $l_\theta \leftarrow \mathcal{L}(\theta; y_t, \hat{y}_t)$ (6);
8      Take a gradient step on $\nabla_\theta l$;
9  **end**

---

---

**Algorithm 2:** Generating from CountsDiff

1 **Input:** number of timesteps $T$; class $c$, guidance strength $\gamma$, attrition schedule $\sigma$;
2 **Output:** $\hat{x}_0$;
3 $x_1 = \vec{0}$;
4 **for** $t_k \in \text{linspace}([1, 0], T)$ **do**
5     $t \leftarrow t_k$ ;
6     $s \leftarrow t_{k-1}$ ;
7     $p(t) \leftarrow \cos^2(\pi t/2)$ ;
8     $p(s) \leftarrow \cos^2(\pi s/2)$ ;
9     $\beta_{t,s} \leftarrow \frac{p(s) - (1 - \sigma_{t,s}) \, p(t)}{1 - p(t)}$;
10    $\hat{y}_{uncond} \leftarrow NN_\theta(x_t, p(t))^+$ ;
11    $\hat{y}_{cond} \leftarrow NN_\theta(x_t, p(t); c)^+$;
12    $\hat{y} \leftarrow (\hat{y}_{cond})^\gamma (\hat{y}_{uncond})^{(1-\gamma)}$;
13    $\hat{y}_{clipped} \leftarrow \text{random\_round}(\hat{y})$;
14    Sample $b_t \sim \text{Bin}(\hat{y}_{clipped}, \beta_{t,s})$;
15    Sample $n_t \sim \text{Bin}(x_t, 1 - \sigma_{t,s})$;
16    $x_s \leftarrow b_t + n_t$ ;
17 **end**

---

## D EXPERIMENTAL SETTINGS

### D.1 SIMULATED COUNTS SETTINGS

Each of the $d = 10$ dimensions is sampled from a negative binomial distribution with parameters $\mu \sim \text{log-uniform}(0.05, 0.5)$ and $\theta \sim \text{log-uniform}(0.2, 5.0)$, which represent the mean and dispersion of the negative binomial, selected log-uniformly from $[0.05, 0.5]$ and $[0.2, 5.0]$ respectively. Each sample is then multiplied by a size factor $s \sim \text{log-normal}(0, 0.6)$ that breaks the independence between dimensions. The parameters were chosen such that the data was sparse ($> 50\%$ zeros) and the max count was sufficiently large ($\approx 50$). The Gaussian diffusion algorithm is DDPM with a cosine noise schedule Dhariwal & Nichol (2021), trained on log-normalized ($y = \log(1 + x)$) counts. Log-normalization is a common pre-processing technique used for biological counts data before downstream analysis. Since absolute errors in log-space correspond to relative errors in count space, this makes the MSE loss in Gaussian diffusion more sensible for the task at hand, where relative errors are more interesting (predicting 99 instead of 100 should not be penalized as much as predicting 1 instead of 2).The discrete diffusion algorithm is taken from Sahoo et al. (2025) with the linear mutual information interpolating schedule recommended in Austin et al. (2021). CountsDiff uses zero death-rate sampling, no guidance, and the continuous, time-inhomogeneous parametrization of the Blackout Diffusion noise schedule.

All three methods were trained with a simple 1-layer multilayer perceptron. Gaussian Diffusion and CountsDiff have 48-dimensional hidden layers, and masked diffusion has a 4-dimensional hidden layer to approximately match the total model weights of the other two models, since the output dimension of masked diffusion is the number of classes, which is $\max(X) + 1 \approx 50$.

All models were trained until convergence, at approximately 4000 gradient steps, $T = 200$ steps were used at sample time. 4000 samples were generated from each model and the joint MMD and SWD and marginal MMD and Wasserstein distance to 4000 samples generated from the ground truth distribution is computed. MMD was computed with the Radial Basis Function (RBF) (Buhmann, 2000) kernel with parameter $\gamma = 1$.

### D.2 NATURAL IMAGES

Because Blackout Diffusion did not release model weights, we elected to re-implement their method using the Unet2D package from Diffusers Huggingface library. Due to GPU memory constraints, we first attempted to reduce batch size, which we found had a significant effect on both model performance and training speed. Instead we elected to halve the number of residual layers per block, which also reduced model performance but converged in the same number of steps as reported in

Santos et al. (2023). The remainder of the model and training hyperparameters were transferred directly from the default CIFAR-10 hyperparameters in Song et al. (2020b), which Blackout Diffusion uses as its base model. We used most of the same hyperparameters for CelebA, except we were forced to reduce the batch size to 100, also consistent with Santos et al. (2023), due to memory constraints. We also slightly adjusted the Unet architecture so that attention is performed at the $16 \times 16$ resolution.

Consistent with Blackout Diffusion, CIFAR-10 models were trained for 300k gradient steps, though it is likely conditional models would have benefited from additional training.

Blackout Diffusion's codebase has CelebA set to train for 1.3 million gradient steps; due to computational constraints, we were forced to terminate training after 600k steps, even though validation metrics were still decreasing.

See code for exact training configs.

### D.3  SCRNA-SEQ IMPUTATION

#### D.3.1  SCRNA-SEQ DATA PREPROCESSING

Given the sparsity and dimensionality of scRNA-seq data, we first filter out genes that are rarely expressed across cells. Only the top 1000 (fetus) and 500 (heart) genes, sorted by coefficient of variance, were selected. We follow Algorithm 1 from (Zhang & Liu, 2024) to identify missingness sites for each sample. We used an 80/10/10 train/val/test split. While evaluating methods on the test set, we used default training settings and tuned sampling hyperparameters on the val set.

#### D.3.2  SCFID IMPLEMENTATION

scRNA-seq transcriptome embeddings were obtained using a pre-trained *Homo sapiens* SCVI model (Lopez et al., 2018) (version **2024-02-12**) from the CZI CELLxGENE Discover platform. SCVI was chosen as the embedding model because it takes raw count data as input. This version was trained on the CELLxGENE Human Census data (release 2023-12-15) and implemented using the scvi-tools package (Gayoso et al., 2022). To score imputations, the scFID score is calculated between the full ground truth expression profiles and the full expression profiles, with the targets replaced by the model predictions.

#### D.3.3  SCRNA-SEQ IMPUTATION BASELINES

To capture a diversity of the existing scRNA-seq imputation methodologies, we compare CountsDiff to MAGIC (Van Dijk et al., 2018), GAIN (Yoon et al., 2018), Hi-VAE (Nazabal et al., 2020), scIDPMs (Zhang & Liu, 2024), ForestDiffusion (Jolicoeur-Martineau et al., 2024), and ReMDM (Wang et al., 2025), along with naive baselines of zero-imputation, mean imputation, and conditional mean imputation (conditioned on sample covariates). All code implementations for scRNA-seq baselines were trained on the same data splits as CountsDiff. Default parameters for models were used where possible.

For GAIN, we adapt training loss to permit fully masked entries. Due to time constraints, we limit training to 50 epochs, after training loss converges. At test time, we provide the entire test set (and corresponding missingness mask) in one shot, enabling GAIN to run normalization over the entire dataset during imputation. Hint matrices were constructed as in the original implementation, with a probability $h = 0.9$ of providing a hint at each locus in the mask.

For Hi-VAE, which only accepts $\mathbb{N}$ for count-based imputation, we alter the model to run on zero counts by incrementing data by one prior to training/imputation and decrementing afterwards. While this enables Hi-VAE to model zeros in the data, it slightly alters the count distribution compared to an alternatuve model with explicit modeling of zeros. Due to time constraints, we trained Hi-VAE for 100 epochs.

For MAGIC, we ran a hyperparameter sweep on the validation set to pick the optimal knn parameter, optimizing over scFID values. At evaluation time, we provide the MAGIC with the entire train and test set to construct the neighbor graph and run our evaluation on the imputed sites for the test set cells only.

For scIDPMs, we consider both single imputation and multiple (5) imputation. While the original scIDPM implementation used multiple imputation, taking the median value item-wise over 100 imputations, we modify this behavior for a fair comparison between generative models in single imputation (CountsDiff, ForestDiffusion, and ReMDM). Due to GPU constraints, we train scIDPMs for 150 hours and use the final checkpoint at evaluation time.

For ForestDiff, we again change the default parameters to perform single imputation. Due to computational constraints, we are unable to report results for ForestDiffusion on the larger fetus cell atlas; ForestDiffusion results are reported for the smaller heart cell atlas in Table E.4.

### D.3.4 DISCUSSION OF SCRNA-SEQ IMPUTATION METRICS

We note that evaluating scRNA-Seq imputation methods is an open problem beyond the scope of this work, and individual metrics often do not tell the full story. To address this, we have included a breadth of metrics, both sample-level and distributional, that quantify different aspects of sample quality. Spearman correlation is a common metric for evaluating imputation that measures the preservation of the relative ordering of the genes sorted by expression. This metric is particularly relevant in downstream tasks where identifying the most highly differentially expressed genes, and not the degree of differential expression, is relevant. RMSE is an error metric that is sensitive to outliers and therefore tends to be higher for models that are more likely to predict unrealistic outliers. Bias is the mean difference between imputed values and real values, and scFID is meant to measure whether the empirical distribution of an imputation method's samples resembles the true empirical distribution.

# E    ADDITIONAL EXPERIMENTS

## E.1    SIMULATED DATA

### E.1.1    REMAINING DIMENSIONS FOR SIMULATED EXPERIMENTS

We consistently observe that CountsDiff and masked diffusion are comparable in joint MMD and marginal metrics, but CountsDiff consistently outperforms masked diffusion on joint-SWD. See figure 7. CountsDiff also consistenly maintains variance closer to (albeit lower than) the real data, whereas Gaussian diffusion collapses, and masked diffusion has much higher variance, indicating the presence of excessive outliers. See table 4.

Table 4: Variance of generated samples per dimension. CountsDiff (Ours) consistently maintains variance closer to the Real Data, whereas Gaussian Diffusion collapses (near-zero variance) and masked diffusion suffers from extreme outliers (high variance).

| Dimension | Real Data (Ground Truth) | CountsDiff (Ours) | Gaussian | Masked |
|---|---|---|---|---|
| Dim 0 | 0.78 | **0.55** | 0.19 | 3.06 |
| Dim 1 | 4.71 | **1.99** | 0.46 | 9.22 |
| Dim 2 | 0.10 | **0.07** | 0.01 | 1.89 |
| Dim 3 | 0.12 | **0.08** | 0.02 | 2.49 |
| Dim 4 | 0.28 | **0.21** | 0.05 | 7.27 |
| Dim 5 | 1.97 | **1.10** | 0.24 | 4.78 |
| Dim 6 | 0.84 | **0.44** | 0.24 | 5.19 |
| Dim 7 | 16.09 | 4.50 | 1.04 | **19.83** |
| Dim 8 | 0.62 | **0.46** | 0.17 | 11.63 |
| Dim 9 | 1.77 | **0.96** | 0.22 | 4.09 |

### E.1.2    PLOTS OF JOINT DISTRIBUTIONS OF TOY DATA

See figure 8

### E.1.3    RANDOMIZED ROUNDING

In the low-counts setting (Figure 9a), we see that the no-rounding method suffers from mode collapse at 0, which fails to preserve the proper marginals. Although both the Poisson approximation and our stochastic random-rounding scheme are empirically effective, in the low-count setting of scRNA-seq data, the Poisson distribution is an unprincipled approximation of a Binomial distribution.

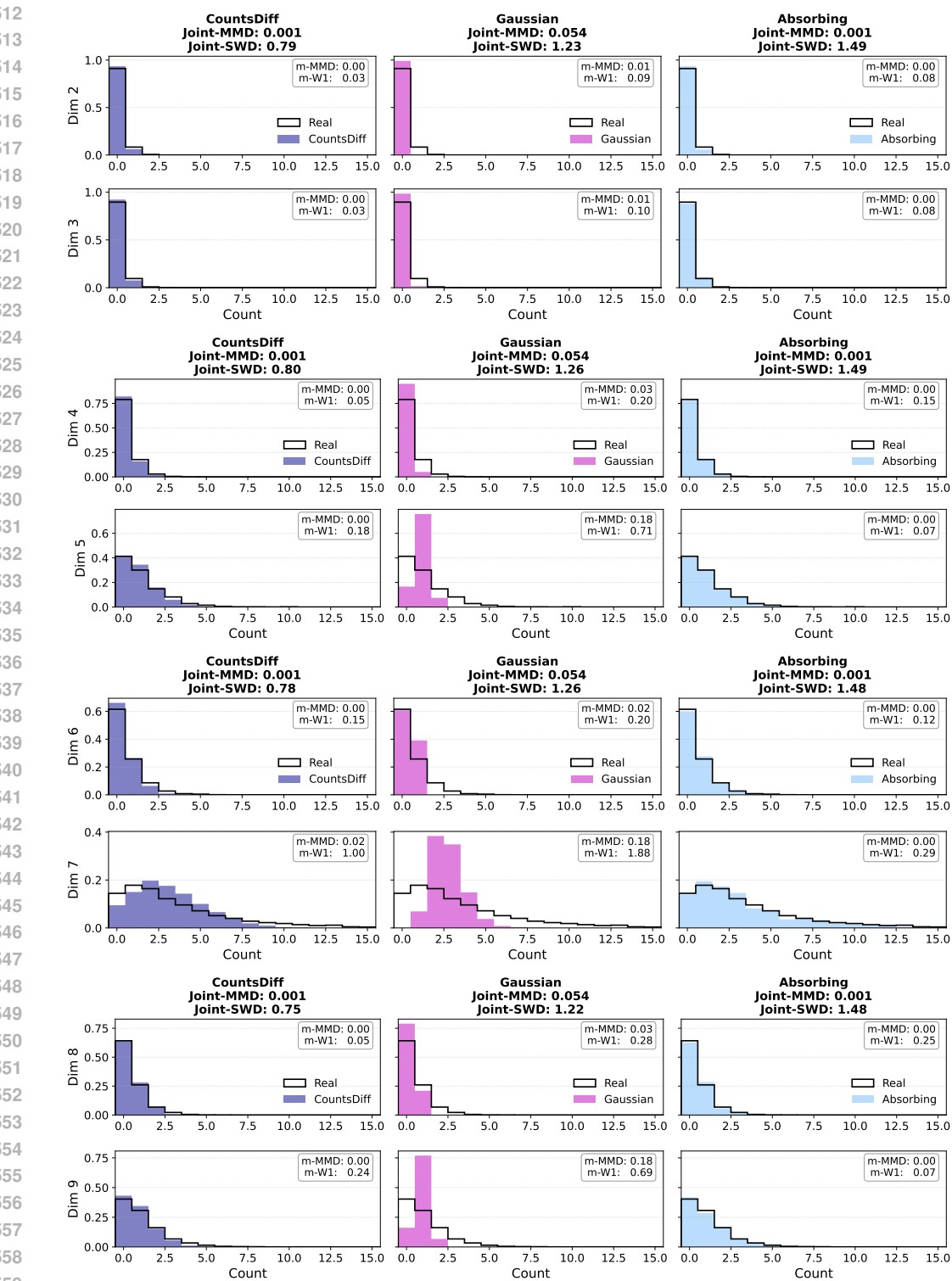

Figure 7: Histograms of marginals of dimensions 2-9

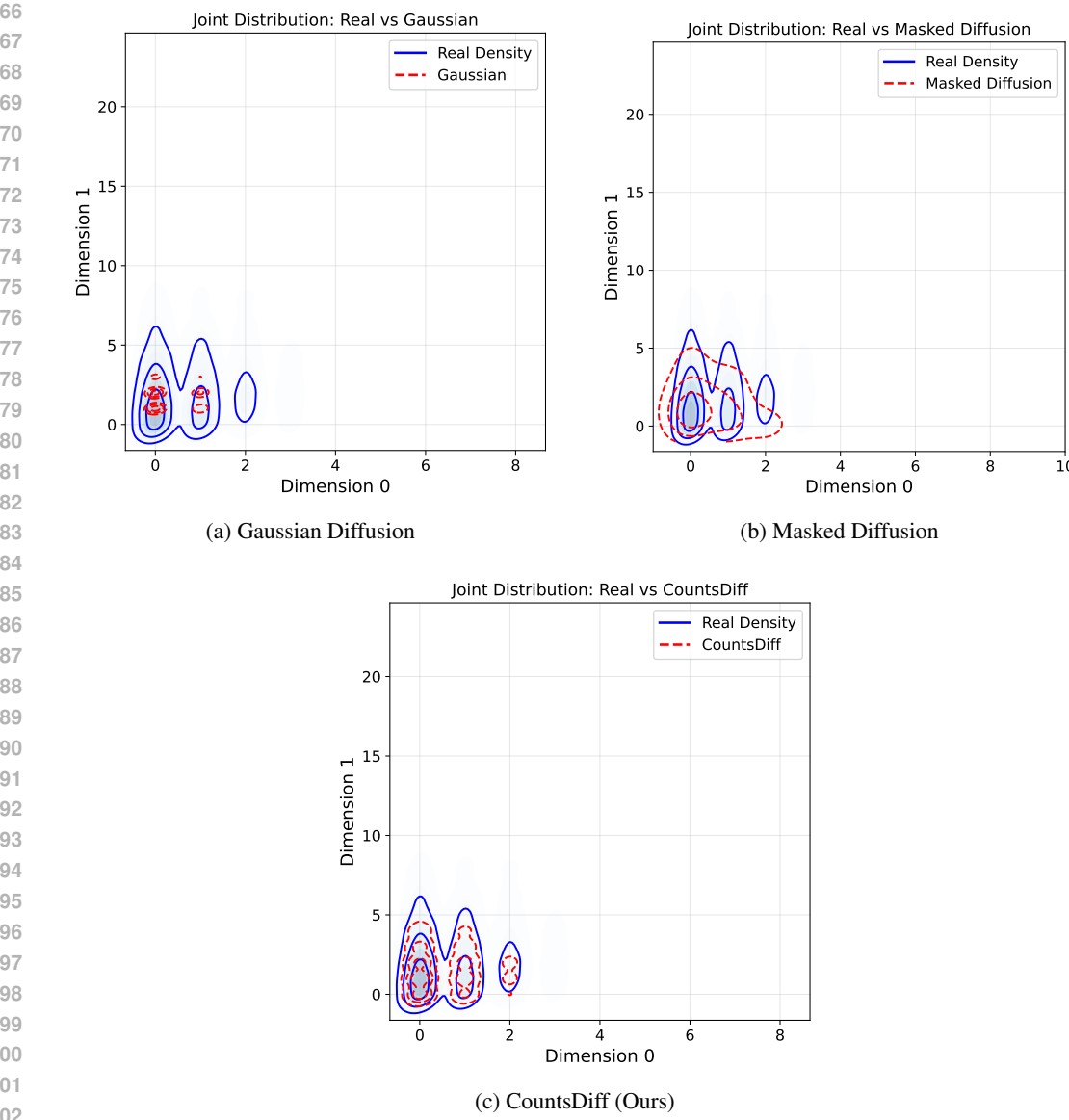

Figure 8: Joint Kernel Density Estimate (KDE) plots between dimensions 0 and 1 of real data (blue contours) versus model-generated samples (red dashed contours). Gaussian Diffusion suffers from mode collapse, resulting in a much tighter, less diverse distribution. Masked Diffusion exhibits a broader, more diffuse distribution with 'leaked' probability mass and slightly less correlation between the dimensions, indicating outliers and overfitting to the marginals. CountsDiff (Ours) more closely aligns with the true data distribution.

## E.2 CIFAR-10

### E.2.1 COSINE $p$-SCHEDULE STABILIZES TRAINING

We observe substantially reduced instability in the training curves of models trained with the cosine noise schedule versus the FI continuous noise schedule, though both seem to converge to the same optimum value for CIFAR-10. See figure 10.

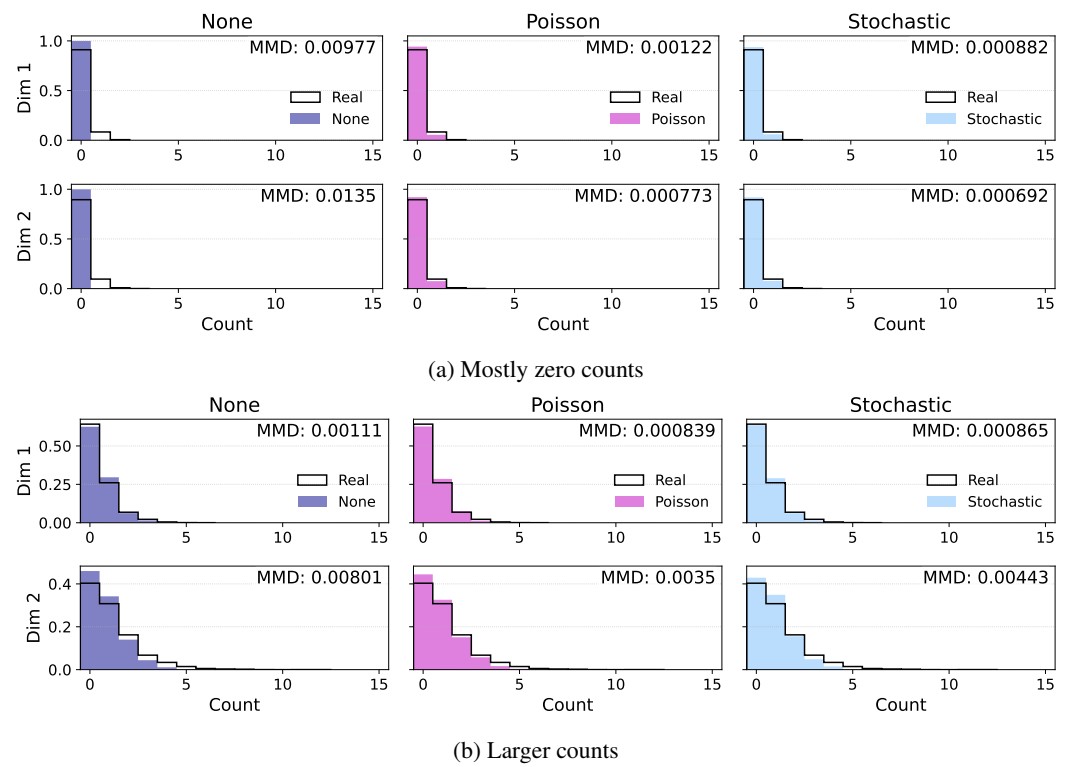

(a) Mostly zero counts

(b) Larger counts

Figure 9: Comparison of Binomial sampling standard rounding, Poisson with no rounding, and Binomial sampling with stochastic rounding across two different counts regimes.

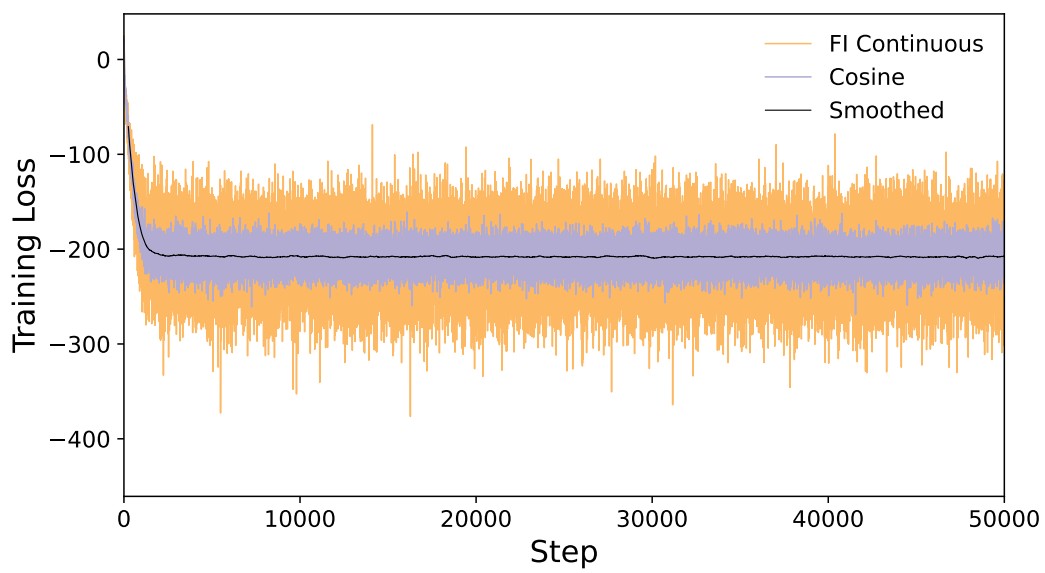

Figure 10: Train loss of CountsDiff over 50k steps for FI continuous and cosine $p$-schedules.

### E.2.2 CIFAR-10 GUIDANCE SWEEP

See Table E.2.2. We observe that at larger guidance scales, increasing guidance improves the IS at the expense of FID, in line with the effect of guidance in other diffusion frameworks. We also

Table 5: FID and IS of 5k images sampled with from CountsDiff with the Continuous FI (left) and Cosine (right) $p$-schedules at various guidance scales with $\eta_{\text{rescale}} = 0.01$

| Guidance Scale | FID $\downarrow$ | IS $\uparrow$ |
|---|---|---|
| 0.0 | 11.648 | $8.537 \pm 0.391$ |
| 0.1 | 11.606 | $8.599 \pm 0.269$ |
| 0.2 | 12.078 | $8.636 \pm 0.460$ |
| 0.5 | 12.145 | $8.943 \pm 0.439$ |
| 1.0 | 9.820 | $9.286 \pm 0.452$ |
| 2.0 | 13.296 | $9.467 \pm 0.342$ |
| 3.0 | 17.620 | $9.337 \pm 0.394$ |

| Guidance Scale | FID $\downarrow$ | IS $\uparrow$ |
|---|---|---|
| 0.0 | 11.233 | $8.952 \pm 0.271$ |
| 0.1 | 11.331 | $8.987 \pm 0.498$ |
| 0.2 | 11.933 | $8.741 \pm 0.396$ |
| 0.5 | 11.985 | $9.001 \pm 0.412$ |
| 1.0 | 9.507 | $9.561 \pm 0.368$ |
| 2.0 | 14.154 | $9.498 \pm 0.370$ |
| 3.0 | 18.542 | $9.641 \pm 0.293$ |
| 5.0 | 24.063 | $9.493 \pm 0.167$ |

find that the cosine $p$-schedule is more responsive to guidance: both metrics vary more in the model trained with the cosine schedule. The cosine schedule also seems to be more stable at moderate guidance scales, while the FI schedule is more stable at extreme guidance scales.

We observe that both the FID and the IS suffer from small guidance scales, indicating poor unconditional generation compared to the unconditional models. We believe this may be caused by training for the same number of iterations as the unconditional models with ap_uncond = 0.2, so the model has effectively one-fourth as many train steps in the unconditional case as the conditional case.

### E.2.3 CIFAR-10 ATTRITION SWEEP

Table 6: FID and IS of 5k images sampled with from CountsDiff various attrition rate strategies

| Attrition Rate Strategy | FID $\downarrow$ | IS $\uparrow$ |
|---|---|---|
| None | 9.666 | $9.463 \pm 0.402$ |
| $\eta_{rescale} = 0.005$ | 9.562 | $9.504 \pm 0.382$ |
| $\eta_{rescale} = 0.01$ | 9.507 | $9.561 \pm 0.368$ |
| $\eta_{rescale} = 0.02$ | 10.400 | $9.653 \pm 0.269$ |
| $\eta_{rescale} = 0.05$ | 12.727 | $9.565 \pm 0.268$ |

Empirically, we find that small, nonzero $\eta_{rescale}$ improved evaluation metrics. We emphasize that these metrics are reported on only $5,000$ samples, so the FID is substantially poorer than it would be for a larger number of samples, as seen in Table 1.

Although the bounds on the attrition schedule for the CountsDiff sampling process resemble those in ReMDM, the strategies that work for remasking in their framework do not necessarily transfer to attrition schedulers in CountsDiff. We hope that future work will shed light on how best to design attrition rate schedules. A particularly exciting direction is value-dependent attrition rates: since the marginal is valid regardless of the attrition rate, one could conceivably set *different attrition rates* for each position in $d$ if, for example, the value in that particular position is deemed "unfit."

## E.3 CELEBA

### E.3.1 QUANTITATIVE METRICS ON CELEBA

Table 7: FID of 10k images sampled with from 30M-parameter CountsDiff model trained on CelebA for 500k steps

| $p$-**schedule** | $\gamma$ | attrition schedule | **FID** $\downarrow$ |
|---|---|---|---|
| FI Continuous | 1.0 | $\eta_{\text{rescale}} = 0.01$ | 9.844 |
| Cosine Continuous | 1.0 | $\eta_{\text{rescale}} = 0.01$ | **7.637** |

Table 8: FID of 50k images sampled with from 60M-parameter CountsDiff model trained on CelebA for 1M steps

| $p$-**schedule** | $\gamma$ | attrition schedule | **FID** $\downarrow$ |
|---|---|---|---|
| Cosine Continuous | 1.5 | $\eta_{\text{rescale}} = 0.005$ | 4.948 |

Table 9: FID of 5k images sampled with from 60M-parameter CountsDiff model trained on CelebA for 1M steps

| $p$-**schedule** | $\gamma$ | attrition schedule | **FID** $\downarrow$ |
|---|---|---|---|
| Cosine Continuous | 1.0 | $\eta_{\text{rescale}} = 0.005$ | 7.580 |
| Cosine Continuous | 1.0 | $\eta_{\text{rescale}} = 0.01$ | 7.541 |
| Cosine Continuous | 1.5 | $\eta_{\text{rescale}} = 0.005$ | 7.217 |
| Cosine Continuous | 1.5 | $\eta_{\text{rescale}} = 0.01$ | 7.483 |

### E.3.2 ATTRITION RATE SMOOTHING

## E.4 HEART CELL IMPUTATION

Table 10: Benchmarking results across evaluation metrics for various models on scRNA-seq imputation on human heart cell atlas with 50% MCAR. Metrics are reported as mean (standard error)

| Model Method | Spearman $\uparrow$ | RMSE$\downarrow$ | Bias | log(scFID) $\downarrow$ |
|---|---|---|---|---|
| RAW | N/A | 9.565(0.643) | $-3.507(0.044)$ | $-0.321(0.006)$ |
| Mean imputation | 0.314(0.001) | 8.542(0.0672) | 0.012(0.023) | $-2.937(0.017)$ |
| Conditional Mean | 0.494(0.001) | 7.536(0.450) | 0.007(0.035) | $-4.205(0.021)$ |
| MAGIC | 0.388(0.003) | 9.392(0.322) | $-3.458(0.043)$ | $-0.379(0.007)$ |
| scIDPMs, 1-sample | 0.312(0.001) | $> 10^3$ | $> 10^3$ | $-1.063(0.009)$ |
| scIDPMs, filtered | 0.312(0.001) | 8.994(0.441) | 1.550(0.010) | $-1.061(0.011)$ |
| scIDPMs, 5-samples | 0.348(0.001) | 6.954(0.194) | 1.035(0.006) | $-3.278(0.007)$ |
| GAIN | 0.240(0.001) | 121.322(0.933) | 6.293(0.105) | $-1.317(0.014)$ |
| Hi-VAE | 0.000(0.001) | 8.832(0.309) | 2.776(0.027) | 0.202(0.005) |
| Forest Diffusion | 0.016(0.001) | 136.218(0.221) | 46.672(0.056) | 0.048(0.007) |
| ReMDM, 1-sample | 0.325(0.002) | 8.877(1.304) | $-\mathbf{0.027(0.010)}$ | $-\mathbf{7.466(0.090)}$ |
| ReMDM, 5-samples | **0.434 (0.002)** | **6.109(0.490)** | $-\mathbf{0.025(0.005)}$ | $-6.509(0.005)$ |
| CountsDiff, 1-sample | 0.312(0.002) | 7.035(0.009) | $-0.160(0.015)$ | $-7.062(0.069)$ |
| CountsDiff, 5-samples | 0.427(0.002) | **6.209(0.624)** | $-0.161(0.012)$ | $-5.814(0.036)$ |

## E.5 CELL CLASSIFIER PREDICTIONS

We trained an XGBoost Classifier with 100 estimators and a max depth of 6 on the training set for both the fetal and heart datasets to predict cell type. Imputed data points were then evaluated

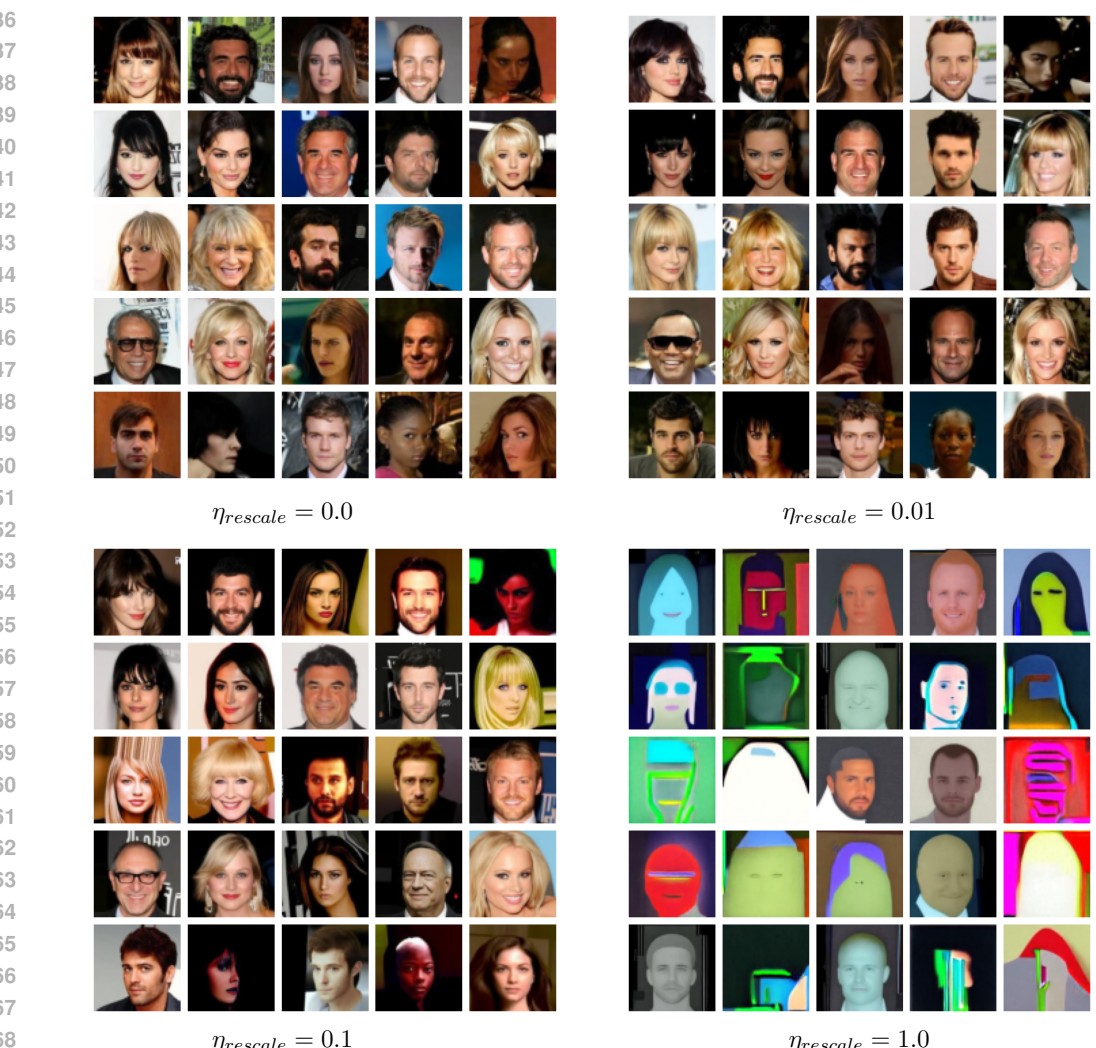

$\eta_{rescale} = 0.0$

$\eta_{rescale} = 0.01$

$\eta_{rescale} = 0.1$

$\eta_{rescale} = 1.0$

Figure 11: 25 images drawn from CountsDiff trained on CelebA with increasing $\eta_{rescale}$.

on classification accuracy and F1-score with the classifier. Using the same imputation missingness schemes, we report the results for the following competitive methods.

Table 11: Cell classifier results for scRNA-seq imputation on human fetal cell atlas with 50% MCAR. Metrics are reported as mean (standard error)

| Model Method | Accuracy ↑ | F1↑ |
|---|---|---|
| RAW | 0.82(0.00) | 0.53(0.01) |
| Mean imputation | 0.81(0.00) | 0.49(0.01) |
| Conditional Mean | 0.82(0.00) | 0.55(0.01) |
| MAGIC | 0.62(0.00) | 0.27(0.01) |
| ReMDM, 1-sample | 0.82(0.00) | 0.53(0.01) |
| ReMDM, 5-samples | 0.82(0.00) | 0.53(0.01) |
| CountsDiff, 1-sample | 0.81(0.00) | 0.50(0.01) |
| CountsDiff, 5-samples | 0.81(0.00) | 0.51(0.01) |

Table 12: Cell classifier results for scRNA-seq imputation on human fetal cell atlas with 25% low-biased MNAR. Metrics are reported as mean (standard error).

| Model Method | Accuracy ↑ | F1↑ |
|---|---|---|
| RAW | 0.82(0.00) | 0.54(0.01) |
| Mean imputation | 0.81(0.00) | 0.51(0.01) |
| Conditional Mean | 0.82(0.00) | 0.53(0.01) |
| MAGIC | 0.76(0.00) | 0.47(0.01) |
| ReMDM, 1-sample | 0.82(0.00) | 0.54(0.01) |
| ReMDM, 5-samples | 0.82(0.00) | 0.54(0.01) |
| CountsDiff, 1-sample | 0.81(0.00) | 0.52(0.01) |
| CountsDiff, 5-samples | 0.82(0.00) | 0.52(0.01) |

Table 13: Cell classifier results for scRNA-seq imputation on human heart cell atlas with 50% MCAR. Metrics are reported as mean (standard error)

| Model Method | Accuracy ↑ | F1↑ |
|---|---|---|
| RAW | 0.99(0.00) | 0.94(0.01) |
| Mean imputation | 0.99(0.00) | 0.93(0.01) |
| Conditional Mean | 0.99(0.00) | 0.95(0.01) |
| MAGIC | 0.93(0.00) | 0.79(0.01) |
| ReMDM, 1-sample | 0.99(0.00) | 0.94(0.01) |
| ReMDM, 5-samples | 0.99(0.00) | 0.940.01) |
| CountsDiff, 1-sample | 0.98(0.00) | 0.94(0.01) |
| CountsDiff, 5-samples | 0.99(0.00) | 0.93(0.01) |

# F  LARGE LANGUAGE MODEL STATEMENT

Large Language Models and associated AI tools were used for the following:

1. Deep research queries to Gemini and ChatGPT were used for retrieval and discovery of related works, to ensure fair credit was given to works we may not have been previously aware of

2. AI IDE assistants were used to aid in debugging, figure generation, and implementation of certain simple, canonical methods.

3. LLM assistants were used intermittently to polish already written text to make it more comprehensible to readers.

