# OpenReview forum: "CountsDiff: A diffusion model on the natural numbers for generation and imputation of count-based data"
_ICLR.cc/2026/Conference — Submitted to ICLR 2026_

### Official Review · Reviewer_Mgjp · 2025-10-31

**Soundness:** 4
**Presentation:** 3
**Contribution:** 2
**Rating:** 4
**Confidence:** 2

**Summary:**

The paper introduces **CountsDiff**, a diffusion framework whose state space is the set of natural numbers. The forward process is a continuous-time pure-death birth–death process parameterized by a _survival_ schedule p(t)p(t)p(t); the reverse process is generalized to allow **attrition** (deaths) during sampling, enabling non-monotone trajectories akin to remasking/churn. The objective is a weighted NLL with a principled time weighting; the authors also add predictor-free guidance and propose randomized rounding to avoid small-count collapse. Experiments cover (i) toy sparse counts, (ii) CIFAR-10/CelebA generation, and (iii) scRNA-seq **imputation** (fetal and heart cell atlases), where CountsDiff is competitive with discrete diffusion (ReMDM/D3PM) and specialized imputation baselines.

**Strengths:**

- *Well written easy and structured:* The forward process and binomial marginals/conditionals are derived cleanly with an intuitive p(t)p(t)p(t) parameterization; the link to Gaussian schedules (via matched SNR) is useful for design transfer.

**Weaknesses:**

- *Novelty*: CountsDiff’s core forward process is a reparameterized pure-death process; the main novelty is the _design space framing_ (survival schedule + weights) and the **attrition** reverse sampler. This is solid but somewhat incremental relative to Blackout diffusion and continuous-time discrete denoising models.
- *Empirical results:* While the aim isn’t SOTA on CIFAR-10/CelebA, reported FIDs are modest and sometimes worse under cosine schedule without tuning; training steps for CelebA are halved due to compute, weakening claims there.

**Questions:**

- Gaussian diffusion on the toy dataset seems to be unexpectedly bad given that the authors were "unable to match the performance ..., even by varying model capacity". Can the authors provide some more details on the setting? And what was used for dequantization (i.e. unfiorm dequnatization, bitwise channel encodings,...)

---

> ### Author Response · Authors · 2025-11-21
> **Response to Reviewer Mgjp**
>
> We thank the reviewer for the thoughtful and detailed review. We are glad that the reviewer finds our work “well written, easy, and structured,” and that they highlight the value of the clean derivation for the forward process, the intuitive parametrization, and the usefulness of the link to Gaussian schedules. We respond to concerns below:
> ### **Novelty and Relationship to prior frameworks**
> **Novelty compared to Blackout Diffusion:**
> To clarify why we consider the contribution substantive, we believe it’s helpful to contextualize the evolution of diffusion modelling design principles:
> Empirical behavior in modern diffusion models is largely attributed to changes in their **design parameters**: noise schedules, loss weightings, prediction targets, reverse-time dynamics (churn/remasking), and guidance. In Gaussian diffusion, this design space was formalized by Karras et al (2022), whose **disentangled design space** **allowed subsequent works to dramatically improve performance**. Similarly, Diffusion Duality (Sahoo et al. 2025\) unifies Gaussian and categorical diffusion from a theoretical perspective, allowing the transfer of design choices. Our work **derives a disentangled design space** and **draws connections to the design space of Gaussian and categorical diffusion**. Blackout diffusion lacks this flexible, deconvolved structure, and is recovered as a special case of CountsDiff with fixed weights, no attrition, no guidance, and discrete-time training:
> Blackout diffusion has:
>
> 1. a fixed p schedule and a loss weighting implied by an observation time schedule.
>    1.  Not obvious how to decouple these design choices and how the observation time schedule corresponds with design choices in Gaussian diffusion.
> 2. No guidance or reverse process modifications
>
> **CountsDiff deconvolves the Blackout diffusion parameters, and generalizes/unifies Blackout Diffusion** by exposing:
>
> 1. **A general p-schedule** with a proven connection to Gaussian noise schedules
> 2. **A principled weighting** with connection to importance sampling
> 3. **A generalized birth-death process** that solves the **fundamental monotonicity limitation**
> 4. **Continuous-time training**, which allows arbitrary sampling steps at test time and enables **predictor-free guidance**
>
> We believe this clarified design space provides the foundation for future advancement in counts generative modeling.
>
> **Novelty compared to continuous-time discrete diffusion frameworks**
>
> Continuous-time formulations, classifier-free guidance, and remasking/churn have indeed been explored in discrete diffusion models, but almost exclusively in the setting of *finite categorical* state spaces (e.g., masked or uniform discrete diffusion). Those models operate on vocabularies of a fixed size without ordering.
>
> While we intentionally frame the design space such that it *resembles* that of existing work to enable seamless transfer of techniques and heuristics, CountsDiff tackles a related but **fundamentally distinct problem** of modeling an ordered but unbounded state space of counts, deriving similar techniques from first principles:
>
> 1. We derive a continuous-time pure-death process on counts whose marginals and conditionals are exactly binomial for arbitrary monotone survival schedules p(t), and the corresponding reverse birth-death process with attrition.
> 2. Thanks to the structure of our state space, we obtain exact continuous-time negative log-likelihood and its associated time weighting, instead of a variational objective with a heuristic weighting schedule.
> 3. We instantiate predictor-free guidance and non-monotone trajectories via attrition, which play a role analogous to guidance and remasking/churn in categorical diffusion, but are derived directly for count-valued processes

---

> ### Author Response · Authors · 2025-11-21
> **Response to Reviewer Mgjp continued**
>
> **Empirical results**
>
> The reviewer correctly notes that the reported FIDs are modest and that the CelebA training was shortened due to compute and time constraints. Since submission, we have acquired additional compute and have trained **larger CountsDiff models** for the full number of steps on CIFAR-10 and CelebA. We have updated the manuscript to include the improved metrics **on the larger models** on CIFAR-10. For CelebA, we have updated appendix 3.1.1 to include some improved quantitative results for a 2x larger CountsDiff model (60M parameters vs 30M) trained for twice as many steps (1M vs 500k). The main conclusions about the relative impact of guidance and attrition remain the same.
>
> We would also like to emphasize:
>
> * We do not claim competitiveness with Gaussian diffusion on natural images; the experiments on CIFAR-10 and CelebA are intended only as a **sanity check** that CountsDiff is capable of modeling high dimensional distributions and to understand the relative effects of our design choices. **We’ve updated section 4 in the manuscript to make this intent clearer.**
> * We are not proposing the cosine schedule as the optimal choice: its purpose is to show that principled Gaussian diffusion design choices can be directly transferred to CountsDiff;
> * The cosine schedule was originally introduced for **stability** in Gaussian diffusion; we observe the same stabilizing effect in CountsDiff (see appendix E.2.1) via a significant reduction in train loss instability.
> * The noise schedule derived from Blackout Diffusion is also a principled schedule based on constant reduction in Fisher Information: CountsDiff’s flexibility simply makes this choice of p-schedule explicit and interchangeable
>
> While CountsDiff is not designed to be SOTA on images, **we have strengthened the empirical results where we do expect it to excel:** we have introduced additional analyses in the toy dataset that illustrate that CountsDiff avoids a key failure case of masked diffusion: over-sampling/hallucinating outliers (see Section 5.1). We observe a similar phenomenon in scRNASeq-imputation, where  “ReMDM's higher RMSE and substantially higher RMSE standard error compared to CountsDiff also reflects masked diffusion's tendency to over-sample outliers, which is particularly undesirable in scientific tasks where these outliers may be misconstrued as signal.” **We believe that this revision much more convincingly underscores CountsDiff’s advantages over existing diffusion models in count-data regimes.**
>
> **Gaussian Diffusion underperforms on toy counts (question 1).**
> We thank the reviewer for prompting clarification here. The full training details for the simulated experiment are **outlined in Appendix D.1**, but we agree that more detail on the preprocessing and dequantization was needed. We do not use uniform or bitwise dequantization; instead we worked directly with log-normalized counts $y = \log(1 + x)$ as continuous inputs to the Gaussian diffusion model. We have revised appendix D.1 to include the following to clarify this point:
> “The Gaussian diffusion baseline is DDPM with a cosine schedule (Dhariwal & Nichol 2021), trained on log-normalized counts $y = \log(1 + x)$. Log normalization is a standard preprocessing step for biological count data, as absolute errors in log-space correspond to relative errors in count space. This makes the MSE loss in Gaussian diffusion more appropriate, since predicting 99 vs. 100 should not incur the same cost as predicting 1 vs. 2.”
> Attached is also the anonymized codebase, which includes the notebook used to produce these toy experiments.
>
> We also would like to give an intuition for why we expect Gaussian diffusion to underperform on these sorts of data: while the underlying distribution defined over the natural numbers is well-behaved, when one extends the support to the real numbers (as is done when modeling counts with Gaussian diffusion), the density becomes highly discontinuous, making it quite difficult to model, particularly when the average number of counts is low.

---

> ### Author Response · Authors · 2025-11-21
> **Response to Reviewer Mgjp (summary and references)**
>
> **Summary**
> The reviewer highlights both the clarity of our framework and its potential for bridging continuous and discrete diffusion design. As we understand, the reviewer’s concerns center on (1) whether CountsDiff is more than an incremental improvement relative to existing work (2) the strength and positioning of empirical results, and (3) the setup of the Gaussian diffusion baseline on toy counts.  We have clarified that CountsDiff **generalizes existing counts diffusion baselines** and provides the foundation for future advances, and **extends continuous-time techniques** to an ordered, unbounded state space. We have also revised the work to include stronger results in regimes where CountsDiff is intended to excel while positioning natural images as a sanity check rather than a target domain. We are happy to answer any additional questions or concerns the reviewer may have.
>
> **References**
>
> 1. Karras, Tero, et al. "Elucidating the design space of diffusion-based generative models." Advances in neural information processing systems 35 (2022): 26565-26577.
> 2. Kingma, Diederik, and Ruiqi Gao. "Understanding diffusion objectives as the elbo with simple data augmentation." Advances in Neural Information Processing Systems 36 (2023): 65484-65516.
> 3. Wang, Guanghan, et al. "Remasking discrete diffusion models with inference-time scaling." arXiv preprint arXiv:2503.00307 (2025).
> 4. Sahoo, Subham Sekhar, et al. "The diffusion duality." arXiv preprint arXiv:2506.10892 (2025).

---

### Official Review · Reviewer_A2oq · 2025-11-01

**Soundness:** 4
**Presentation:** 3
**Contribution:** 3
**Rating:** 6
**Confidence:** 4

**Summary:**

In this paper, the authors develop a continuous-time diffusion framework (CountsDiff) that is suited for modeling natural numbers (count data). Specifically, the proposed framework extends a prior model, Blackout diffusion, by introducing features adapted from modern continuous diffusion models, such as continuous-time training, classifier-free guidance, and non-monotone reverse dynamics (attrition). The authors validate the framework on natural images and RNA sequences, showing it can match or outperform a SOTA discrete generative model .

**Strengths:**

1. The formulation of a pure death process as the forward corruption process and a generalized birth-death process (attrition) as the reverse sampling process provides an elegant and principled way of modeling distributions on the natural numbers.

2. The proposed framework successfully bridges the gap between modern Gaussian-based continuous diffusion models and discrete diffusion models in count-based domains, specifically by translating and incorporating powerful techniques such as continuous-time training, classifier-free guidance, and non-monotone reverse dynamics. This greatly unlocks the modeling potential of diffusion models for count data.

**Weaknesses:**

1. The experiments are relatively weak and somewhat unconvincing. Despite all the technical innovations, the experimental results of CountDiff do not fully demonstrate its advantages. For example:

   i. On natural images, there are no other baselines except for the variants of CountDiff itself. It is difficult to tell how well the proposed method performs without a direct comparison to other methods under the same setting (*e.g.*, same model architecture and training steps).

   ii. In Table 1, it seems that to achieve good sampling results (FID and IS), both guidance $\gamma$ and attrition schedule $\eta_{\text{rescale}}$ have to be tuned. It would be helpful to include ablation results showing the individual effect by just changing one of them.

2. The paper fails to mention some highly-relevant prior work [1]. Upon a quick research, it appears that Chen et al. (2023) [1] developed a similar discrete-time forward and backward processes that are suited for modeling count data, which the authors termed binomial thinning and Poisson thickening processes respectively. In [1], an identical loss function was introduced to train the model. However, it does not explicitly acknowledge nor discuss the established foundation laid by this prior work.

[1] Chen, Tianqi, and Mingyuan Zhou. "Learning to jump: Thinning and thickening latent counts for generative modeling." International Conference on Machine Learning. PMLR, 2023.

**Questions:**

1. Although the paper formulates CountDiff as a continuous-time diffusion framework for count data. It is unclear what the additional benefits are to adopt the continuous-time training. Unlike the Gaussian diffusion, to my understanding, CountDiff still have to go through a fixed number sampling steps as described by Algorithm 2. Consequently, the major drawback of slow sampling remains unresolved in this new framework.

2. Can the CountDiff be adapted for modeling non-negative continuous data like [1]? In [1], a trick named the Poisson-based data randomization is introduced to handle both natural numbers and non-negative real values.

3. The main quantitative comparison (Table 1) reports the best FID and IS for different combinations of guidance ($\gamma$) and attrition ($\eta_{\text{rescale}}$). Could the authors clarify how to determine a good combination of the two hyperparameters in practice?

---

> ### Author Response · Authors · 2025-11-21
> **Response to Reviewer A2oq**
>
> We appreciate this careful and in-depth review. We are glad that the reviewer finds our work “**elegant and principled”**, and we appreciate their recognition that the proposed design space “**greatly unlocks the modeling potential of diffusion models for count data.**” We would like to address their concerns in detail below.
>
> ### **Comparison to Chen & Zhou (2023) “Learning to Jump”**
>
> We thank the reviewer for bringing this work to our attention. We have revised the related works section to reflect its contributions and limitations in relation to CountsDiff, copied below for reference
> “JUMP(Chen & Zhou (2023)) models positive, real-valued data by projecting it into counts ($z\_0 \sim \mathrm{poisson}(\lambda x_0)$), then noises through a binomial thinning of $z_0$, resulting in a similar noising process, parametrized by $\alpha_t$. Their loss objective, derived from the ELBO, resembles our objective but with a different predictive target and constant loss weighting. For natively count-based data, JUMP(Chen & Zhou (2023)) also propose Binomial-JUMP, which can be interpreted as another special case of CountsDiff with constant weights, no guidance, and no attrition, and uses the Poisson sampling scheme mentioned in section 3.5. JUMP's primary advantage lies in its ability to handle continuous non-negative data, which is outside the scope of the present work. The underlying noising and denoising resembles Blackout Diffusion and has a similarly limited design space. JUMP and CountsDiff are complementary approaches, and CountsDiff's improvements on modeling counts (extended design space, continuous-time formulation, and exact loss) can be readily extended to non-negative reals using JUMP's Poisson data randomization trick (Appendix A.5).”
>
>  We believe that this resolves the missing relevant work identified by the reviewer.
>
> ### **Comparison to baselines, ablations, and experimental results**
>
> (i) Lack of natural image baselines:
>
> We would like to note that we are using natural images as a **sanity check** to:
>
> 1. Stress-test the scalability to high-dimensional data distributions (see intro: line 64; 66 in revision)
> 2. Probe the *relative effects* of the design parameters we introduce (schedules, weighting, guidance, attrition) in a domain that is both visually interpretable and has well-defined benchmarks (see section 4, line 267; 269 in revision)
>
> Therefore, **we do not intend to directly compete with generative models designed for image generation** and we do not expect SOTA-level experimental results in images. While more baselines on natural images may better contextualize the performance on images, we focus on results on variants of CountsDiff in order to align with the objectives of our experiments in natural images. We have revised section 4 to more clearly communicate the objectives of these experiments on natural images.
> We would also like to clarify that **FI-Discrete with no guidance** in table 1 corresponds exactly to Blackout Diffusion, our baseline. Our manuscript has been revised to reflect the above clarification.
>
> (ii) Ablations of guidance and attrition: We appreciate the reviewer pointing out the difficulty of locating the ablation study in our manuscript. After revisiting the manuscript, we discovered a **typo** that incorrectly directed the reader to Appendix E.3 (CelebA) as opposed to E2.2 and E2.3 which contain the requested ablations. We have corrected this typo in the updated manuscript.
>
> We also provide the full grid (all combinations of guidance and attrition) in the attached codebase under data/evals/big-sweep/, in addition to the full grid of the newly trained **larger CountsDiff models** (4 residual blocks per layer with more training steps) under data/evals/big-sweep\_4layer. As mentioned in  note that due to the cost of generating and evaluating 50k samples for each set of hyperparameters, metrics are computed for 5k samples, resulting in reduced absolute FID performance (though we find the relative performance between hyperparameter choices is stable).
>
> We would also like to mention that we have introduced additional analyses in the toy dataset that illustrate that **CountsDiff avoids a key failure case of masked diffusion**: over-sampling/hallucinating outliers (see Section 5.1). We observe a similar phenomenon in scRNASeq-imputation, and  “ReMDM's higher RMSE and substantially higher RMSE standard error compared to CountsDiff also reflects masked diffusion's tendency to over-sample outliers, which is particularly undesirable in scientific tasks where these outliers may be misconstrued as signal.” We believe that **this revision much more convincingly underscores CountsDiff’s advantages over existing diffusion models in count-data regimes.**

---

> > ### Author Response · Authors · 2025-11-21
> > **Response to Reviewer A2oq Continued**
> >
> > ### **Choosing Guidance and Attrition hyperparameters**
> >
> > As mentioned in the previous point, the guidance and attrition hyperparameters were chosen with a sweep over guidance scale and a one-parameter family of attrition schedules borrowed from Remasking Diffusion Models (Wang et al., 2025). Fortunately, the direct connection to hyperparameters in existing diffusion models makes the search space intuitive:
> >
> > * The optimal guidance scale is comparable to values used on classifier/predictor-free guidance in Gaussian and discrete diffusion models
> > * The optimal $\eta$-parameter for the  $\eta$-rescale schedule matches the order of magnitude of the optimal $\eta$ reported in ReMDM.
> >
> > These connections allow for practitioners to utilize familiar heuristics from existing literature rather than an overly broad hyperparameter sweep.
> >
> > ### **Advantages of continuous-time training**
> >
> > Although our sampling procedure still requires time discretization, adopting continuous-time training provides several meaningful advantages:
> >
> > 1. Sampling steps can be chosen freely *at test time*: in discrete time training, one must set a series of time-steps, which then must be used at inference time. In contrast, learning a continuous function of time allows for the number of steps to be a sampling hyperparameter. This allowed us to discover that scRNA-seq sample quality, for example, saturates at \~50 steps, dramatically improving sampling speed.
> > 2. Formulating the reverse process in continuous time in terms of transition rates allows for the ratios that enable guidance to emerge naturally
> > 3. Compatibility with fast discrete-diffusion ODE/SDE solvers: when attrition rate is set to 0, we have a closed form for the transition rates in terms of the model outputs, making it directly compatible with recently developed high-order solvers (see Ren et al. 2025), which we plan to explore in follow-up work.
> > 4. Empirical improvement in sample quality: Table 1 shows that FI-Continuous outperforms FI-discrete without any other changes.
> >
> > ### **Extension to non-negative continuous data**
> >
> > While our work focuses on native count data (which allows us to preserve the exact NLL), the Poisson-based data randomization trick from Chen & Zhou (2023) can indeed be combined with CountsDiff via the following procedure:
> >
> > 1. Nonnegative real inputs $x_0$ are mapped to latent counts via $z_0 \sim \mathrm{Poisson}(\lambda x_0)$ (Poisson-based data randomization)
> > 2. CountsDiff is applied directly to model the distribution of $z_0$
> > 3. Generated samples are divided by $\lambda$ at inference time (original distribution of x\_0 is recovered as $\lambda \rightarrow \infty$, shown in Chen & Zhou (2023))
> >
> > This simple procedure would extend all the benefits of CountsDiff (guidance, schedule design, loss weighting, and attrition) to continuous, non-negative domains. As mentioned in the previous point, our continuous time formulation also in principle unlocks fast ODE/SDE solvers for JUMP. It would not, however, be a *strict* generalization of JUMP, as the model will operate directly in the latent counts space, and predict $z_0 - z_t$ as opposed to $x_0$ in JUMP. We have revised the manuscript to mention this generalization in the related works with further detail in appendix A.5.
> >
> > **Summary**
> >
> > As we understand the reviewer’s concerns, they center on (1) adding more direct baselines on natural images, (2) clarifying the ablations for guidance and attrition, (3) explicitly acknowledging prior work, and (4) explaining the benefits of continuous-time training and the selection of hyperparameters in practice. We have revised the manuscript to address each of these points, corrected the appendix pointer and substantially expanded the related-work section. **We have also added simulated experiments that demonstrate CountsDiff’s advantages over masked diffusion in counts data generation and note a similar advantage in real-world data.** We hope these clarifications fully resolve any concerns, and we thank the reviewer again for their positive and constructive assessment of the core framework.
> >
> > **References**
> >
> > 1. Chen, Tianqi, and Mingyuan Zhou. "Learning to jump: Thinning and thickening latent counts for generative modeling." International Conference on Machine Learning. PMLR, 2023\.
> > 2. Wang, Guanghan, et al. "Remasking discrete diffusion models with inference-time scaling." arXiv preprint arXiv:2503.00307 (2025).

---

### Official Review · Reviewer_Pqn8 · 2025-11-01

**Soundness:** 2
**Presentation:** 3
**Contribution:** 2
**Rating:** 4
**Confidence:** 2

**Summary:**

This paper presents a discrete diffusion model that builds off of Blackout diffusion with a reparameterization of the forward process, which enables the large toolkit from Gaussian diffusion (for reals) to be applied in the natural number case. The corresponding denoising process (with generalization over birth-death processes), guidance, and rounding procedures are worked out. On simulated counts, discrete diffusion methods easily capture the data distribution where Gaussian diffusion fails. On CIFAR10, countsdiff produces competitive qualitative and quantitative results. Finally, the authors test the method on imputation for transcriptomics data.

**Strengths:**

I will contextualize both my positive and critical comments by acknowledging that I’m not very well versed in the details of existing discrete diffusion methods, and so may not be able to judge the conceptual advances (or the lack thereof) of the current work:

- the paper tackles an important problem in developing diffusion models more appropriate for discrete data with an elegant and more general framework building off of existing works.
- the results are solid, and while not overwhelmingly superior than existing methods, provide some interesting insights on the method, such as the effect of attrition scale on blurring and the newly incorporated noise schedules
- the paper is overall clearly written with concise language, and clearly motivates the problem being tackled relative to existing solutions

**Weaknesses:**

- While the framework developed here is elegant, it’s ultimately unclear to me how well it performs compared to existing approaches without the additional proposed generalization. See questions for specific comments. To be clear, I’m not saying that this work is inferior or uninteresting because it does not beat the SOTA on those benchmarks, but that it’s unclear to me as a relatively naive reader what the value of the proposed generalization is aside from conceptual elegance (which I appreciate)

**Questions:**

- It’s great that the method was validated on simulated data, but it’s somewhat expected to outperform a Gaussian model and that’s not really the interesting comparison here. Are there some synthetic experiments (e.g., with higher dimensional toy data) one can come up with that better emphasize the benefits of the generalization compared to existing discrete diffusion methods?

- Similarly, while qualitative results on CIFAR10 look fine and show the value of guidance, attrition, and a cosine p-schedule, it’s unclear to me how they compare quantitively to existing methods (esp. in Table 1). This may be solved, for example, by simply noting that one of the previous methods (e.g. Blackout) is equivalent to which noise schedule.

- On the RNA seq data as well, the current method is comparable but not clearly superior to existing methods, in particular remasking diffusion. The authors reason why some metrics are less meaningful. But as a non-domain expert, a more explicit demonstration of these points (e.g., via a downstream task) may help the authors’ case.

---

> ### Author Response · Authors · 2025-11-21
> **Response to Reviewer Pqn8**
>
> We thank the reviewer for the in-depth review. We are grateful that they highlighted (1) the importance of developing diffusion models tailored to discrete data, (2) the elegance and clarity of our framework, (3) the solid empirical results and insights (e.g., the role of attrition and new schedules), and (4) the clear motivation relative to existing solutions. We also appreciate that the reviewer mentions that they are not deeply familiar with existing discrete diffusion methods; we have kept this in mind and hope we can clarify the value of our contributions as well as provide some context. We address the reviewer’s comments and questions below.
>
> ### **Clarifying the value of the generalization**
>
> The reviewer’s concern centers on understanding the practical value of the generalization beyond conceptual elegance. We hope the following can clarify this:
> Modern diffusion models derive much of their empirical performance from design choices: noise schedules, loss weightings, prediction targets, and reverse-time modifications such as churn or remasking. Empirically, even subtle changes to these parameters can dramatically affect sample quality (e.g., Karras et al. 2022; Kingma & Gao 2023). However, these design choices are deeply entangled in existing count-based diffusion methods. For example, in Blackout Diffusion:
>
> * the noise schedule is fixed,
> * the loss weighting is implicit,
> * and the reverse process is restricted to monotone trajectories.
>
> These constraints can make hyperparameter optimization more challenging and less intuitive, and the resulting progress on counts data diffusions slow. CountsDiff addresses this by providing a deconvolved and interpretable design space:
>
> * Noise schedule p(t), with direct analogue to Gaussian diffusion noise schedule
> * Loss weighting w(t), with link to weighting in Gaussian diffusion and importance sampling
> * Reverse-process dynamics via the attrition rate, which **addresses a** **fundamental theoretical shortcoming** of Blackout Diffusion (analogous to failure-to-remask in masked diffusion (Wang et al., 2025)), namely that the monotonic reverse process does not allow for correction if the model overshoots.
> * Continuous-time training and sampling: allows for number of sampling steps to be an **inference-time hyperparameter** which potentially dramatically improves sampling time and unlocks guidance
> * Guidance scale: conditional generation is necessary for most applications of generative models
>
> This design space is analogous to the unification provided by Karras et al. (2022) for Gaussian diffusion, whose disentanglement of the continuous diffusion design space directly enabled significant empirical gains in later works. We’ve also intentionally parametrized these design parameters to map one-to-one to design parameters in existing diffusion modeling frameworks, similar to Diffusion Duality (Sahoo et al. 2025), which showed that bridging categorical and continuous diffusion unlocks new modeling possibilities by transferring effective design principles across frameworks.
> Our primary aim is analogous: to provide a disentangled framework for count diffusion that allows for the direct transfer of principled design choices from continuous and discrete diffusion to ordinal data. Our example is the cosine schedule, which improves stability in Gaussian diffusion and stabilizes CountsDiff. We believe this design space is essential to unlock improvements in  future work: even if the immediate instantiation is not SOTA on every benchmark, it lays the groundwork for systematic improvement in the domain of generative modeling of counts data.
>
> ### **Additional synthetic experiments**
> Thanks for this excellent question, and we appreciate the reviewer’s interest in the specific advantages of CountsDiff over categorical diffusion models. We’ve done additional analysis on the synthetic data, namely adding Wasserstein-based metrics, analyzing the per-dimension variance of samples, and plotting approximate joint distributions of dimensions with KDE plots in generated samples. In doing so, we’ve uncovered failure cases of masked diffusion, discussed in section 5.1 and appendix E.1 accordingly. We’ve summarized these updates below as well:
>
> 1. Masked diffusion is comparable in marginal metrics and joint MMD to CountsDiff, but performs worse in joint sliced Wasserstein-1 distance (SWD). This is due to:
>    1. **Masked diffusion being prone to generating excessive outliers/hallucinations.** Confirmed by substantially higher variance in masked diffusion samples
>    2. **Masked diffusion also seems slightly less effective at learning relationships between dimensions.** Confirmed visually via KDE-plots

---

> > ### Author Response · Authors · 2025-11-21
> > **Response to Reviewer Pqn8 Continued**
> >
> > In the scRNASeq-imputation results (section 5.3), we have also revised the manuscript to note that “ReMDM's higher RMSE and substantially higher RMSE standard error compared to CountsDiff also reflects masked diffusion's tendency to over-sample outliers, which is particularly undesirable in scientific tasks where these outliers may be misconstrued as signal.” We believe that **this revision underscores CountsDiff’s advantages over categorical diffusion models in simulated and real scenarios.**
> >
> > ### **Comparison to prior methods on CIFAR-10**
> >
> > We thank the reviewer for pointing out the need for clearer comparative interpretation. We have now clarified in the manuscript that **Blackout Diffusion corresponds exactly to the Fisher Information (FI) schedule in discrete time with no guidance** (see updated table 1).
> >
> > ### **scRNA-seq imputation and downstream evaluation (Question 3\)**
> >
> > We appreciate the reviewer for pointing out that non-domain experts may find it difficult to interpret metrics such as Spearman correlation in MNAR settings. As the reviewer suggests, a downstream task can provide clearer evidence. To this end, we train a downstream cell-type classifier on the imputed values of the most competitive methods (CountsDiff, ReMDM, MAGIC), which we have included in appendix E.5. We find that CountsDiff and ReMDM perform comparably on both datasets, and both perform better than other competitive methods.
> > However, we would like to note the limitations of this simple downstream cell-type classifier as an evaluation method:
> >
> > 1. Competitive methods impute *conditioned on cell type and other labels*, so ReMDM’s undesirable tendency to memorize outliers in the training data is *beneficial* for this task.
> > 2. Cell-type and many other common labels can usually be predicted very well by very few genes, so better classification accuracy does not necessarily imply higher-quality samples.
> >
> > Evaluating scRNA-Seq imputation methods is an open problem beyond the scope of this work. In light of this, to aid in reader understanding, we have included a breadth of metrics, both sample-level and distributional, that **quantify different aspects of sample quality**, such as relative ordering of predicted gene expression (Spearman), tendency to predict unrealistic outliers (RMSE), sample bias, and approximate distributional deviation from true expression levels (scFID). **We’ve revised the manuscript to add Appendix D.3.4, which discusses these points.**
> > As mentioned above, we have also revised the results section (5.3) to discuss **the implications of higher versus lower RMSE**. We hope these comments and our revisions shed some additional light on the interpretation of the metrics.
> >
> > ### **Summary**
> >
> > As we understand the reviewer’s concerns, they center on (1) the practical value of our generalization, (2) the need for additional results on simulated data (3) quantitative comparison on CIFAR-10, and (4) clarity on metrics and stronger empirical verification on scRNA-seq imputation. We have detailed the value of the generalization, **revised the work to highlight a major failure case of masked diffusion in simulated data and scRNA-seq imputation that CountsDiff avoids** and clarified the baselines in our CIFAR-10 experiments. We’ve also added a discussion on each of the metrics for those less familiar with the field. We are happy to address any additional questions or concerns should the reviewer have any.
> >
> > ## **References**
> >
> > 1. Karras, Tero, et al. "Elucidating the design space of diffusion-based generative models." Advances in neural information processing systems 35 (2022): 26565-26577.
> > 2. Kingma, Diederik, and Ruiqi Gao. "Understanding diffusion objectives as the elbo with simple data augmentation." Advances in Neural Information Processing Systems 36 (2023): 65484-65516.
> > 3. Wang, Guanghan, et al. "Remasking discrete diffusion models with inference-time scaling." arXiv preprint arXiv:2503.00307 (2025).
> > 4. Sahoo, Subham Sekhar, et al. "The diffusion duality." arXiv preprint arXiv:2506.10892 (2025).

---

### Official Review · Reviewer_Dd4F · 2025-11-02

**Soundness:** 2
**Presentation:** 2
**Contribution:** 2
**Rating:** 2
**Confidence:** 2

**Summary:**

This paper presents CountsDiff, a diffusion framework designed to model distributions on the natural numbers. It builds upon the Blackout Diffusion framework, simplifying its formulation through a direct parameterization in terms of a survival probability schedule and an explicit loss weighting.

**Strengths:**

- Introduces a diffusion framework designed to handle discrete ordinal data using birth/death processes.

- The survival-probability-based parameterization provides a potentially simpler formulation than prior discrete diffusion frameworks.

- The method could be useful for applications such as modeling scRNA-seq data, where count-valued distributions naturally arise.

**Weaknesses:**

- The motivation for applying the method to natural image datasets is unclear and misaligned with the model’s intended domain.

- The novelty relative to existing discrete diffusion models (e.g., Santos et al., 2023) appears limited.

- The presentation and related work discussion are difficult to follow and lack clear differentiation from prior methods.

**Questions:**

I do not understand why CountsDiff is applied to natural image datasets (CIFAR-10, CelebA). Existing Gaussian diffusion models already perform very well, and there is no comparison with these established methods on image datasets. The DDPM paper by Ho et al. (2020) achieved an FID score of 3.17, which is much better than the results reported in Table 1.

The scRNA-seq data seem to make much more sense for this framework, and the authors also mainly emphasize related work on scRNA-seq in Section 6. Therefore, I think the paper should be significantly revised to make this emphasis much clearer, so that the audience can better understand the main contribution. Moreover, the related work section is unclear—for example, the authors mention that methods such as Forest-Diffusion can also be adapted to the scRNA-seq task, but it is not explained what the limitations of that method are or what advantages the proposed approach offers. In addition, Forest-Diffusion is not included as a baseline in the numerical results.

The presentation of the proposed method and its novelty compared to existing work such as Santos et al. (2023) are also unclear. It seems that the main difference from Santos et al. (2023) is that this work considers a general schedule p(t), whereas Santos et al. (2023) focuses on the special case of a fixed p(t).

---

> ### Author Response · Authors · 2025-11-21
> **Response to Reviewer Dd4F**
>
> e thank the reviewer for their thoughtful and detailed feedback. We are glad that CountsDiff’s potential for modeling scRNA-seq data was recognized. We would like to clarify some possible misconceptions and address the reviewer’s concerns below.
>
> Regarding the motivation, before delving into the details, we would like to clear up a misunderstanding at the outset: our motivation is biological counts data, such as scRNA-seq (see lines 44-45), and our experiments on natural images were included only as **sanity checks**. We will elaborate more on this point later.
>
> Regarding related work/novelty relative to Santos et al.:
>
> **Clarifying the contributions relative to Santos et al.**
> This review states that our work “simplifies” Blackout diffusion (Santos et al.), and that the “main difference” is the generalization of p(t). While we are glad our method is judged simpler, our contribution is not to simplify but rather **extend Blackout Diffusion**.  Blackout Diffusion is recovered as a special case of CountsDiff with a fixed $p-$schedule, implicit loss weighting, a monotone reverse process with no attrition, and no guidance or conditional generation. CountsDiff generalizes *each of these axes* in order to elucidate a flexible, interpretable design space for diffusion models on count data, analogous to Karras et al. (2022) for Gaussian diffusion.
>
> Concretely, **CountsDiff makes the following major extensions beyond Blackout Diffusion:**
>
> 1. A general CTMC construction with arbitrary p-schedules:
>    We derive *inhomogeneous* pure-death processes and the corresponding reverse rates that yield closed-form marginals and exact loss objectives for *any* differentiable monotone schedule *p(t),* not just $e^{-t}$ This reparametrization directly links $p(t)$ to Gaussian noise schedules via matching the log-SNR, enabling the transfer of Gaussian diffusion schedule designs
> 2. A disentangled design space :
>    CountsDiff exposes and decouples key design parameters that are either fixed, entangled, or not present in Blackout:
>    1. Noise schedule *p(t)* (fixed in blackout)
>    2. Loss weighting *w(t)* (implicitly defined in Blackout)
>    3. Reverse-process dynamics through attrition rate (not present in Blackout)
>    4. Guidance scale (not present in Blackout)
>
>    Elucidating this design space and connecting it to Gaussian diffusion mirrors the role of Karras et al. (2022) and Diffusion Duality (Sahoo et al., 2025\) which elucidates the design space of Gaussian diffusion models and bridges the gap between Gaussian and Categorical diffusion respectively. These works and ours, which unify the forms of diffusion modeling, are crucial in allowing them to advance together.
>
> 3. A non-monotonic sampling family via attrition:
>    CountsDiff’s generalization of the sampling process addresses a **fundamental theoretical shortcoming** of Blackout Diffusion (analogous to failure-to-remask in masked diffusion (Wang et al., 2025)), namely that the monotonic reverse process does not allow for correction if the model overshoots. Our attrition parametrization is chosen to closely match remasking in discrete diffusion, again to enable the transfer of knowledge across diffusion frameworks
>
> **Clarification about the purpose of CIFAR-10 and CelebA experiments**
> As mentioned above, we tested CountsDiff on images as a **sanity-check** and a **diagnostic tool**. We are not advocating using discrete diffusions for image generation; as correctly noted, Gaussian diffusion models are a more appropriate tool and their significantly better FID on CIFAR-10 (\<= 3.17) than CountsDiff is fully expected. Our reason to include this sanity-check is to:
>
> 1. Stress-test the scalability to high-dimensional data distributions (see intro: line 64; line 66 in revision)
> 2. Probe the *relative effects* of the design parameters we introduce (schedules, weighting, guidance, attrition) in a domain that is visually interpretable with well-defined benchmarks (see section 4, line 267; line 269 in revision)
>
> We realize that we did not make it sufficiently clear that image generation was a diagnostic exercise, not an application target, so **we have revised Section 4 accordingly to avoid the impression that we are proposing our method as a contender for image generation**. We thank the reviewer for pointing this out.

---

> > ### Author Response · Authors · 2025-11-21
> > **Response to Reviewer Dd4F continued**
> >
> > **Focus on scRNA-seq data and improved positioning relative to existing work**
> > As mentioned above, and agreed by the reviewer, scRNA-seq is the natural domain for our framework: in fact, this was the primary motivation of the work (see lines 44-45). Our strongest results are indeed on the biological count data, and we emphasize the results on scRNA-seq in our discussion. Following the reviewer’s guidance, we have **revised the related works section** to make this emphasis clearer by separating the work related to our theoretical contributions and the work related to our primary application (scRNA-seq imputation). We hope that this better explains CountsDiff’s position both in the realm of diffusion modeling and in the imputation of biological counts data.
> >
> > We’ve also **revised the manuscript to add results from Forest-Diffusion.** We did not include it initially because it is primarily designed for tabular/low-dimensional structured data, and is not optimized for high-dimensional count data. However, we agree omitting it leaves a gap, so we have added Forest-Diffusion results in this revision (tables 2-3). We note that (1) Forest-Diffusion scales less efficiently data dimensionality (resulting in 2 orders of magnitude longer wall clock time) and (2) the inclusion does not change any conclusions drawn from the work: it performs poorly compared to CountsDiff in the scRNA-seq setting.
> >
> > **Summary**
> > As we understand, the reviewer’s concerns center on (1) the purpose of our experiments on natural images, (2) novelty relative to prior methods, and (3) unclear presentation of the positioning of our work. We hope to have clarified these points and clarified any misunderstandings. We have also restructured section 4 (experiments) and section 6 (related works) to more clearly motivate the experiments on natural images and clarify CountsDiff’s positioning and novelty relative to related works. We are happy to answer any additional questions or concerns that the reviewer may have.
> > **References**
> >
> > 1. Karras, Tero, et al. "Elucidating the design space of diffusion-based generative models." Advances in neural information processing systems 35 (2022): 26565-26577.
> > 2. Sahoo, Subham Sekhar, et al. "The diffusion duality." arXiv preprint arXiv:2506.10892 (2025).
> > 3. Wang, Guanghan, et al. "Remasking discrete diffusion models with inference-time scaling." arXiv preprint arXiv:2503.00307 (2025).

---

### Author Response · Authors · 2025-11-21
**Code availibility**

An anonymized version of our codebase is available here:
https://anonymous.4open.science/r/countsdiff-7582/README.md

The repository includes instruction and a script (download_data.sh) to retrieve the dataset and model checkpoints needed to reproduce results.
Please reach out through OpenReview if you encounter any issues accessing or running the code.

---

### Meta-Review · Area_Chair_WyS9 · 2026-01-07

**Summary:**

This paper proposes a diffusion framework whose state space is the set of natural numbers. As noted by reviewers Dd4F and Mgjp, the novelty of this work appears limited. As Reviewer Dd4F further notes, the motivation for applying the method to natural image datasets remains unclear.  In addition, the discussion of related work is insufficient, and the empirical improvements over existing methods are marginal, as highlighted by Reviewers Dd4F, Pqn8, and A2oq. Based on these limitations, I agree with the reviewers and recommend rejection.

**Reviewer Concerns:**

Some questions on clirification have been addressed, such as the question raised by reviewer Mgjp. Concerns stated in summary are still outstanding.

**Reviewer Scores:**

There is no clear evidence to suggest that any reviewer would have changed their score after a full discussion.

---

### Decision · Program_Chairs · 2026-01-26

Reject